



# S3M 5.1: a distributed cryospheric model with dry and wet snow, data assimilation, glacier mass balance, and debris-driven melt

Francesco Avanzi[1], Simone Gabellani[1], Fabio Delogu[1], Francesco Silvestro[1], Edoardo Cremonese[2], Umberto Morra di Cella[2, 1], Sara Ratto[3], and Hervé Stevenin[3]

[1]CIMA Research Foundation, Via Armando Magliotto 2, 17100 Savona, Italy
[2]Climate Change Unit, Environmental Protection Agency of Aosta Valley, Loc. La Maladière, 48-11020 Saint-Christophe, Italy
[3]Regione Autonoma Valle d'Aosta, Centro funzionale regionale, Via Promis 2/a, 11100 Aosta, Italy

**Correspondence:** Francesco Avanzi (francesco.avanzi@cimafoundation.org)

**Abstract.**

By shifting winter precipitation into summer freshet, the cryosphere supports life across the world. The sensitivity of this shifting mechanism to climate, as well as the role played by the cryosphere in the Earth energy budget, has motivated the development of a broad spectrum of predictive models. Such models rarely combine a high degree of physical realism in both the seasonal snow and glaciers, and generally are not integrated with hydrologic models describing the fate of meltwater through the hydrologic budget. We present S3M v5.1, a spatially explicit and hydrology-oriented cryospheric model that successfully reconstructs seasonal snow and glacier evolution through time and that can be natively coupled with distributed hydrologic models. Model physics include precipitation-phase partitioning, snow and glacier energy and mass balances, snow rheology and hydraulics, and a data-assimilation protocol. Comparatively novel aspects of S3M with respect to the existing literature are an explicit representation of the spatial patterns of snow liquid-water content, an hybrid approach to snowmelt that decouples the radiation- and temperature-driven contributions, the implementation of the Δh parametrization for distributed ice-thickness change, and the inclusion of a distributed debris-driven melt factor. Focusing on its operational implementation in the Italian north-western Alps, we show that S3M provides robust predictions of the snow and glacier mass balances at multiple scales, thus delivering the necessary information to support real-world hydrologic operations. S3M is well suited for both operational flood forecasting and basic research, including future scenarios of the fate of the cryosphere and water supply in a warming climate. The model is open source, and the paper comprises an user manual as well as resources to prepare input data and set up computational environments and libraries.





## 1 Introduction

The cryosphere is a decisive driver of the Earth system (Barry, 2011; Beniston et al., 2018). Besides altering surface albedo and so concurring to the regulation of global temperature (Flanner et al., 2011), snow and glaciers accumulate winter precipitation and release it during the warm, summer season, when demand is comparatively high (Barnett et al., 2005). This shift in water supply supports water, food, and energy security across climates (Viviroli et al., 2007), with key implications for worldwide societies and ecosystem services (Sturm et al., 2017). For example, snow represents up to 80% of annual water supply in the

semi-arid, largely summer-dry western US (Bales et al., 2006; Serreze et al., 1999; Skiles et al., 2018), while 1.4+ billion people in Asia rely on discharge from high-mountain, cryosphere-dominated regions (Immerzeel et al., 2010). Meanwhile, the Andean cryosphere represents a significant freshwater resource for semi-arid regions of South America (Masiokas et al., 2020), with an estimated contribution of up to 27% to dry-season water supply in La Paz, Bolivia (Soruco et al., 2015).

Seasonality between winter accumulation and summer melt, compounded by equally complex but more short-term pro-
cesses such as rain-on-snow (Rössler et al., 2014), challenges decision makers like water-resources or hydropower managers, who need early and diverse information about snow-glacier amount, distribution, and melt timing to make accurate decisions on water use, allocation, and storage (Georgakakos et al., 2004; Anghileri et al., 2016; Avanzi et al., 2018). This need has catalyzed the development of a large portfolio of models to predict snowmelt- and icemelt-driven discharge (DeWalle and Rango, 2011), to the extent that cryosphere modeling is a dominating topic of both basic and applied contemporary geo-
sciences (Dozier et al., 2016). Application for cryosphere models are not limited to water-supply and flood forecasting, but include avalanche forecasting (Bartelt and Lehning, 2002; Vionnet et al., 2012), land-surface and so weather modeling (Dutra et al., 2010; Wang et al., 2017), and snowmaking (Hanzer et al., 2020). The projected rise in future temperature and aridity (IPCC, 2013) further prioritizes robust predictions of cryospheric water resources, because shrinking glaciers and decreasing snow accumulation may endanger water supply and its predictability (Harrison and Bales, 2016) – especially during droughts
(Huning and AghaKouchak, 2020).

Cryospheric models intersect hydrology with thermodynamics and rheology and as such present a bewildering variety in process representation. Regarding seasonal snow, options range from detailed, physics-based micro-scale models like the Swiss SNOWPACK (Bartelt and Lehning, 2002) or the French Crocus (Vionnet et al., 2012), to intermediate-complexity, simplified-energy-balance models like the UEB model (Tarboton and Luce, 1996) and SNOBAL (Marks et al., 1998), or to simple
one-layer, temperature-index models (Martinec, 1975; De Michele et al., 2013; Avanzi et al., 2015). From a glacier standpoint, the most recurring distinction resides around glacier movement being captured through complex ice-flow approaches, glacier-specific parametrizations of changes in thickness (Huss et al., 2010), an equilibrium relationship between glacier area and long-term climate (Schaefli et al., 2007), or non-dynamic mass balance (Bongio et al., 2016). Parametrizing melt beneath supraglacial debris is another frequent aspect of modeling discretion (Fyffe et al., 2014).

While detail in process representation may appear as the prime driver of model selection, in hydrologic practice this choice also depends on other four pragmatic factors, which make hydrology-oriented cryospheric models essentially different from those oriented to, e.g., avalanche forecasting. First, streamflow generation in cold regions involves not only snow and glacier



ice, but also precipitation-topography interactions (Blanchet et al., 2009; Mott et al., 2014; Cui et al., 2020), vegetation-water feedback mechanisms (Zheng et al., 2016; Avanzi et al., 2020b), and soil-water storage (Bales et al., 2011). Thus, snow-

ice hydrology is inherently spatially distributed and multi-scale (Dozier et al., 2016), with the focus being arguably more on distributed and on point predictions (Blöschl, 1999). Second, high-elevation cryospheric regions remain largely ungauged (Rasmussen et al., 2012; Avanzi et al., 2020a), meaning the necessary input data to run complex models is sparse. This condition has favored parsimonious models (Bartolini et al., 2011) and data-assimilation schemes to remedy model deficiencies with independently observed data (Andreadis and Lettenmaier, 2006; Piazzi et al., 2018). Coupled with data sparsity is the third

factor, that is, the evidence that simplified and complex models often yield comparable predictive accuracy for processes relevant to the seasonal freshet, such as Snow and Ice Water Equivalent (see DeWalle and Rango, 2011, for a definition), surface melt, and runoff (Huss et al., 2010; Avanzi et al., 2016; Magnusson et al., 2015). This explains why hydrology-oriented cryospheric models tend to have low complexity when it comes to internal layering and micro-scale properties. Fourth, processes relevant to cryosphere water resources span horizons from a few hours (such as rain-on-snow events, see Würzer

et al., 2017) to decades (such as glacier dynamics, see Huss et al., 2010), implying that models used for real-world forecasting must be efficient enough to provide landscape-scale predictions in a timely manner (say, a few hours, see Pagano et al., 2014). Ultimately, these four factors trace back to empiricism rather than reductionism being the dominant (and perhaps most successful) paradigm in hydrology (Savenije, 2009), owing to unresolved issues related to upscaling mechanicistic laws to the landscape and measuring the complete heterogeneity of hydrologic processes (Blöschl and Sivapalan, 1995; Beven, 2006).

Here, we present Snow Multidata Mapping and Modeling (S3M) v5.1, a snow and glacier model developed and maintained by CIMA Research Foundation (https://www.cimafoundation.org/). S3M fulfills all the four factors of hydrology-oriented cryospheric models outlined above, including being spatially distributed, parsimonious as for both input-data requirements and complexity, and enough computationally efficient to be deployed in operational, real-time flood-forecasting chains (Laiolo et al., 2014). Specific aspects of interest in S3M are (1) a spatially explicit prediction of both dry and wet-snow spatial patterns

and so bulk snowpack liquid water content ($\theta_W$, in vol%), an increasingly decisive variable for snowmelt and avalanche forecasting (Techel and Pielmeier, 2011; Wever et al., 2014; Avanzi et al., 2015; Wever et al., 2016); (2) the combination of both snow and glacier mass dynamics in a coherent modeling framework, including the so-called $\Delta h$ parametrization by Huss et al. (2010) and melt beneath supraglacial debris; (3) provisions for assimilating various decision-relevant variables like SWE, snow depth, and satellite-based snow-cover area. S3M v5.1 is the last generation of a model originally proposed by Boni et al.

(2010), but significantly developed thereafter (henceforth, simply S3M).

The paper is organized as follows: Section 2 focuses on model description, including both snow and glaciers. Section 3 presents an example of results for an inner alpine valley in north-western Italy where various versions of this model have been operational since the early 2000s (Aosta valley). Finally, sections 4 discusses model applicability and future developments. The Appendix includes an User Manual discussing run preparation, execution, and post-processing.





## 2  Model description

S3M is a raster-based model, with the same set of equations being solved for each cell and no spatial interdependency besides glacier change in thickness. A planned future release will include interdependency in the form of wind redistribution and melt routing through the snowpack (see Section 4). The time step of the model is flexible, but it is generally set to 1 hour. All input, state, and output variables are *distributed*, meaning they are passed to the model as rasters with fixed resolution in geographic degrees (see the Appendix). All equations are solved for all pixels in the simulation domain using a forward-Euler method.

### 2.1  Definitions

We define snow and glacier ice as a mixture of three constituents: ice, liquid water, and air. Following De Michele et al. (2013) and Avanzi et al. (2015), the control volume of unitary area ($h_{TOT}$) for each pixel of the simulation domain is defined as:

$$h_{TOT} = h_G + h_S = \frac{V_{tot}}{A} \tag{1}$$

where $h_G$ is glacier thickness (m), $h_S$ is the height of snow (often referred to as snow depth, m), $V_{tot}$ is the control volume in m$^3$ and $A$ is the area of the pixel; $h_G = V_G/A$ and $h_S = V_S/A$, where $V_G$ and $V_S$ are the total volume of glacier and snow within the pixel under study. We define $M_S$ and $M_G$ as the mass of seasonal snow and glacier ice for each pixel, respectively (in kg). As for seasonal snow, this mass is $M_S = M_D + M_W$, with $M_D$ and $M_W$ the mass of the dry (snow grains) and wet (interstitial liquid water) constituents, respectively. The mass of air is assumed to be negligible compared to $M_D$ and $M_W$ (De Michele et al., 2013).

Snow is a foam of ice (Kirchner et al., 2001), meaning the dry constituent occupies a porous skeleton of height $h_D$ (m, volume $V_D$) and porosity $n = h_P/h_D$ (-), where $h_P$ is the height of pores in the mixture (m, volume $V_P$). Thus, we define the density of the solid skeleton $\rho_D$ (in kg m$^{-3}$) as:

$$\rho_D = \frac{M_D}{V_D} = \frac{\rho_i(V_D - V_P)}{V_D} = \rho_i(1 - n), \tag{2}$$

where $\rho_i = 917$ kg m$^{-3}$ is ice density. The density of the wet constituent is equal to that of liquid water: $\rho_W = M_W/V_W = 1000$ kg m$^{-3}$ ($V_W$ is the volume of liquid water in the mixture, height $h_W$). The density of glacier ice is assumed equal to $\rho_i$.

During most of the snow season, the dry and wet constituents of the snowpack are both contained in $h_D$, the prevalent volume. However, an oversaturation condition takes place during the last instants of the snow season due to phase change (De Michele et al., 2013). Despite being a limit, virtually unmeasureable scenario, including this oversaturation condition is important from a numerical-stability standpoint. Thus, the total control volume of snow $h_S$ (that is, snow depth) is hereby defined as:

$$h_S = h_D + \langle h_W - n h_D \rangle, \tag{3}$$





where $\langle\rangle$ are Macaulay brackets, which provide the argument if this is positive, otherwise 0. In other words, $h_S$ is equal to the height of the porous structure $h_D$ plus – if present – the oversaturated volume $\langle h_W - nh_D \rangle$ (De Michele et al., 2013).

Accordingly, the bulk snow density $\rho_S$ (in kg m$^{-3}$) is

$$\rho_S = \frac{M_D + M_W}{V_S} = \frac{\rho_D V_D + \rho_W V_W}{V_S} = \frac{\rho_D h_D + \rho_W h_W}{h_S} \tag{4}$$

and Snow Water Equivalent (in m w.e.) is $SWE = \rho_S \times h_S \times \rho_W^{-1}$. We also define the bulk volumetric liquid water content of the snowpack as:

$$\theta_W = \frac{V_W}{V_S} = \frac{h_W}{h_S}. \tag{5}$$

Glaciers are modeled as a single-phase material, the mass of liquid water and air being being negligible compared to glacier ice. Thus, the Ice Water Equivalent for glaciers ($IWE$) is:

$$IWE = \frac{\rho_i h_G}{\rho_W}. \tag{6}$$

Figure 1 summarizes the main definitions, state variables, and inputs of S3M.

## 2.2 Snow: mass-conservation equations

The mass-conservation equations for the dry and wet constituents of the snowpack read as follows:

$$\frac{dM_D}{dt} = \hat{S}_f - \hat{M} + \hat{R} \tag{7}$$

$$\frac{dM_W}{dt} = \hat{R}_f + \hat{M} - \hat{R} - \hat{O}, \tag{8}$$

where $\hat{S}_f$ is the snowfall mass flux, $\hat{M}$ is the snowmelt mass flux, $\hat{R}$ is the refreezing mass flux, $\hat{R}_f$ is the rainfall mass flux, and $\hat{O}$ is the outflow mass flux (also known as snowpack runoff, see Avanzi et al., 2019). Given that $M_D = \rho_D h_D A$, we

simplify Equation 7 as

$$\frac{1}{\rho_W A} \frac{d(\rho_D h_D A)}{dt} = \frac{\hat{S}_f - \hat{M} + \hat{R}}{\rho_W A} \tag{9}$$

to obtain

$$\frac{d}{dt}\left(\frac{\rho_D h_D}{\rho_W}\right) = \frac{dSWE_D}{dt} = S_f - M + R, \tag{10}$$

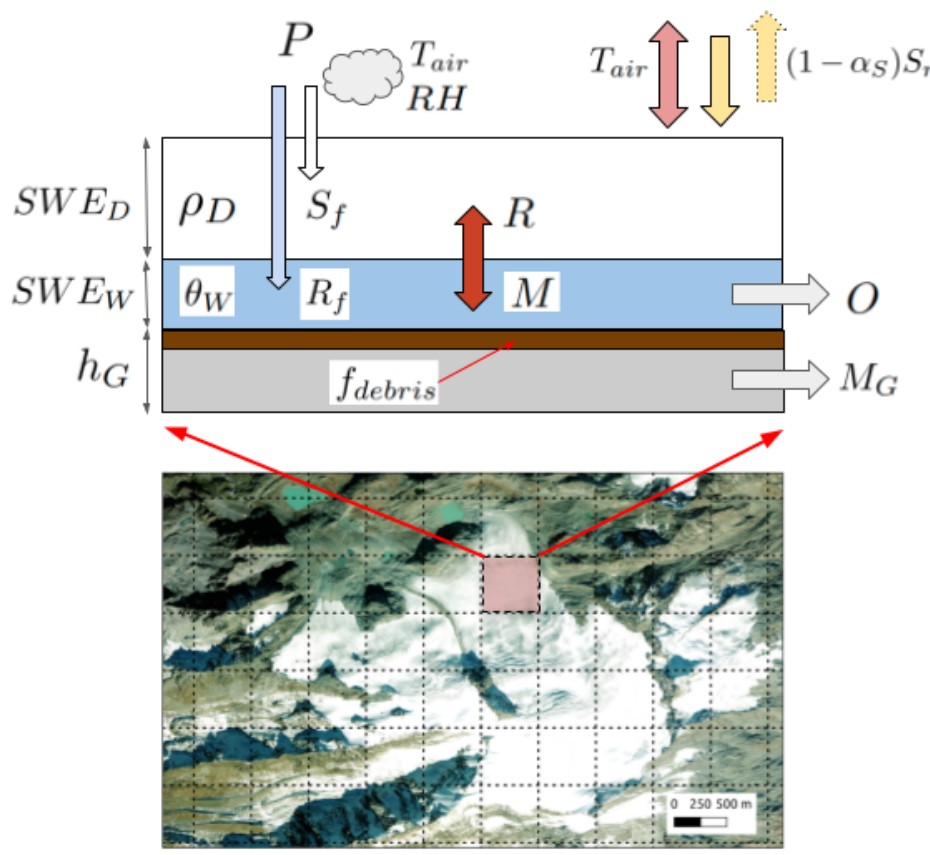

**Figure 1.** Main definitions, state variables, and inputs of S3M (see Section 2 for details). $P$ is total precipitation, $T_{air}$ is air temperature, $\alpha_S$ is snow albedo, $S_r$ is incoming shortwave radiation, $S_f$ and $R_f$ are snowfall and rainfall rate, respectively, $RH$ is relative humidity, $SWE_D$ and $\rho_D$ are dry Snow Water Equivalent and dry bulk snow density, respectively, $SWE_W$ and $\theta_W$ are wet Snow Water Equivalent and bulk volumetric liquid water content, $R, M$, and $O$ are snow refreezing, melt, and outflow, respectively, $h_G$ is glacier thickness, $f_{debris}$ is a coefficient accounting for the modulating effect of thick debris on ice melting, $M_G$ is ice melt. The background image is Rutor glacier in north-western Italy.

where $SWE_D$ is the dry-snow water equivalent ($\rho_D h_D \rho_W^{-1}$) and $S_f$, $M$, and $R$ are the snowfall, snowmelt, and refreezing
mass fluxes in mm w.e. $\Delta t^{-1}$. Note that $SWE_D$ and all related mass fluxes will henceforth be expressed in mm w.e., with a conversion by 1000 mm m$^{-1}$ being implicitly included between Equation 7 and 10. Likewise, we simplify Equation 8 as

$$\frac{1}{\rho_W A} \frac{d(\rho_W h_W A)}{dt} = \frac{\hat{R}_f + \hat{M} - \hat{R} - \hat{O}}{\rho_W A} \tag{11}$$





to obtain

$$\frac{dSWE_W}{dt} = R_f + M - R - O, \tag{12}$$

with $SWE_W$ and $R_f$ and $O$ being the rainfall and snowpack-runoff mass flux in mm w.e (again, note that $SWE_W$ and all related mass fluxes will henceforth be expressed in mm w.e., with a conversion by 1000 mm m$^{-1}$ being implicitly included between Equation 8 and 12).

Equations 10 and 12 are the two fundamental mass-conservation equations of S3M, which thus offers a spatially explicit, prognostic simulation of both the dry and wet constituents of snow. This phase separation follows De Michele et al. (2013) 145    and Avanzi et al. (2015), with two differences. First, equations in De Michele et al. (2013) and Avanzi et al. (2015) were written using $h_D$ and $h_W$ as main state variables, whereas here we used $SWE_D$ and $SWE_W$, which allows a more compact formulation of mass-conservation equations since no density-compaction term is necessary in Equation 10. Second, Avanzi et al. (2015) introduced a mass-conservation equation for a third constituent, refrozen ice. Here, we directly included the refreezing term in Equations 10 and 12.

## 2.3    Snow: mass-flux parametrizations

Mass fluxes in Equations 10 and 12 requiring specific parametrizations are snowfall and rainfall ($R_f$ and $S_f$), snowmelt and refreezing ($M$ and $R$), and snowpack runoff ($O$).

### 2.3.1    Precipitation-phase partitioning

Snowfall and rainfall in S3M are estimated from total precipitation ($P$), an input for the model; precipitation-phase partitioning 155    is based on the empirical approach described in Froidurot et al. (2014):

$$S_f = p_s P \tag{13a}$$

$$R_f = p_r P \tag{13b}$$

$$p_f = 1 - p_r \tag{13c}$$

$$p_r = \frac{1}{1 + e^{\alpha + \beta T_{air} + \gamma RH}}, \tag{13d}$$

where $p_s$ and $p_r$ are the probabilities of snowfall and rainfall, respectively, $\alpha$, $\beta$, and $\gamma$ are fixed parameters derived by Froidurot et al. (2014), $T_{air}$ is air temperature in °C, and $RH$ is relative humidity in %. S3M assumes $p_s$ and $p_r$ to be equal to the actual proportions of rainfall and snowfall over total precipitation. Following Froidurot et al. (2014) and references therein, $\alpha = 22$, $\beta = -2.7$, and $\gamma = -0.2$.





Fresh snow is assumed to be dry, with density $\rho_f$ depending on air temperature (Pomeroy and Brun, 2001):

$$\rho_f = 67.9 + 51.25 e^{\frac{T_{air}}{2.59}}. \tag{14}$$

### 2.3.2 Snowmelt and refreezing

Snowmelt ($M$) is computed if *both* concurrent $T_{air}$ *and* mean air-temperature over the previous 10 days ($\bar{T}_{10d}$) are greater than, or equal to, $T_\tau$, a user-defined threshold usually assumed equal to 1 °C (Pellicciotti et al., 2005); otherwise, $M = 0$. The first condition is standard in degree-day models and accounts for snow melt occurring during periods with a supposedly positive energy balance (meaning a net gain of energy for the snowpack). The second condition is a novel addition of S3M to the literature to keep track of cold content using a pragmatic and parsimonious approach (cold content being a measure of the snowpack-energy deficit to be satisfied for actual melt to start, see Jennings et al., 2018). The basic idea is that $\bar{T}_{10d}$ evolves with a certain delay compared to $T_{air}$, so that setting an additional threshold on $\bar{T}_{10d}$ helps avoiding non-physical melt during short warm spells that come after a somewhat long cold period. Other simple approaches to estimate cold content exist, such as that based on mean-seasonal temperature by Schaefli and Huss (2011), but are all in their early stages. Our approach is the result of intensive trial and error, and seems yielding satisfactory results – especially in suppressing mid-winter melt episodes that do not appear in validation data.

Snowmelt is parametrized using a hybrid physics-based and degree-day approach decoupling radiative forcing from temperature-driven melt (similarly to Pellicciotti et al., 2005):

$$M = m_{rad} \left[ \frac{(1 - \alpha_S) S_r}{\rho_W \lambda_f} \right] \Delta t + c_M m_r (T_{air} - T_\tau), \tag{15}$$

where $S_r$ is incoming shortwave radiation (an input for S3M), $\alpha_S$ is snow broadband albedo, $\lambda_f$ is the specific latent heat of fusion (0.334 MJ kg$^{-1}$), $m_r$ is a degree-day parameter, $m_{rad}$ is a modulating factor to convert shortwave radiation into actual melt (similar to an efficiency parameter, see below), and $c_M$ is a timestep-adjusting parameter.

S3M assumes $S_r$ to be in W m$^{-2}$, $T_{air}$ in °C, $m_{rad}$ to be adimensional, and $m_r$ in mm °C$^{-1}$ day$^{-1}$. While $S_r$ is internally converted to MJ m$^{-2}$ according to model timestep (see $\Delta t$ in Equation 15), the air-temperature part of Equation 15 is computed by first considering an equivalent day with average temperature equal to $T_{air}$, regardless of modelling time step. The snowmelt part depending on air temperature is then readjusted by $c_M = \Delta t / 86400$ to pass from one day to the actual time step ($\Delta t$ is in seconds). This workaround is due to the degree-day approach being originally conceived for daily applications (see for instance Hock, 1999, 2003; De Michele et al., 2013, and references therein). Note that S3M internally sets $S_r$ to 0 between 7PM and 7AM according to forcing timestamps, while proper unit conversions are implicitly included in the radiation part of Equation 15 to first pass from J m$^{-2}$ to MJ m$^{-2}$ and then from m w.e. to mm w.e.

Broadband albedo is computed once per day at midnight (timestamp time) according to Laramie and Schaake (1972):

$$\alpha_S = 0.5 + 0.45 e^{-\tau_\alpha A_s}, \tag{16}$$





where $\tau_\alpha$ is an albedo-decay coefficient equal to 0.12 d$^{-1}$ if average air temperature over the previous 24 hours is higher
than 0°C and equal to 0.05 d$^{-1}$ otherwise. $A_s$ is snow age (in d), defined as the number of days since the last significant
snowfall. S3M considers as significant snowfall one day with at least 3 mm of total snowfall. Snow age $A_s$ is updated every
day at midnight (timestamp time): if cumulative snowfall during the previous 24 hours is less than 3 mm, then snow age is
increased by 1 day; if not, than snow age is reset to 0.

Similarly to other snowmelt models (Rango and Martinec, 1995), melt parameters $m_r$ (mm °C$^{-1}$ d$^{-1}$) and $m_{rad}$ (-) are
calibration-based, although the sensitivity of S3M to both is rather low. This low sensitivity is likely because explicitly sep-
arating the radiation- and temperature-driven components of melt brings these parameters closer to physics than standard
degree-day approaches (see Section 3). Yet, this hybrid approach used by S3M has been rarely considered in the literature.
For example, the seminal work by Hock (1999) did include potential radiation, but embedded it into a degree-day parameter
rather than explicitly separating the radiation and the temperature components. Pellicciotti et al. (2005) are among the few
examples where the radiation and the temperature component are decoupled, but they focused on glacier ice during summer,
which is an isothermal, very efficient condition for shortwave radiation to convert into actual melt. However, applying the orig-
inal approach by Pellicciotti et al. (2005) to snow revealed a tendency to overestimate melt rate early in the snowmelt season,
because it assumes net shortwave radiation to translate into melt regardless of the actual cold content. In subfreezing condi-
tions, in fact, a fraction of net shortwave radiation is used to raise snow temperature, a mechanism that becomes increasingly
unimportant as the season progresses and snow conditions tend towards isothermal. To mimic this transition from subfreez-
ing to isothermal conditions, we propose a modification to the original approach by Pellicciotti et al. (2005) in the form of a
10-day-temperature-modulated efficiency parameter $m_{rad}$ that increases with $\bar{T}_{10d}$ according to a sigmoid function (Figure 2,
a):

$$m_{rad} = 0.49338 \times \arctan(0.27439 \times \bar{T}_{10d} - 0.5988) - 0.49338 \times \frac{3.14}{2} + m'_{rad}. \tag{17}$$

While parameter $m'_{rad}$ is user-defined, we note that $m'_{rad} \sim 1.10$ means that $m_{rad} \to 1$ when $\bar{T}_{10d} > 10$ °C. This corre-
sponds to a 1:1 conversion of net shortwave radiation into melt when isothermal conditions like those by Pellicciotti et al.
(2005) dominate. On the other hand, $m_{rad} \to 0$ when $\bar{T}_{10d} \to 0$ °C. $m_{rad}$ is set to 0 if the equation above predicted a negative
value.

S3M considers a similar relation between $\bar{T}_{10d}$ and $m_r$ through a sigmoid function and a user-defined tuning parameter ($m'_r$,
see Figure 2, b):

$$m_r = 0.598862 \times \arctan(0.27439 \times \bar{T}_{10d} - 0.5988) - 0.598862 \times \frac{3.14}{2} + m'_r. \tag{18}$$

Here again, we note that $m'_r \sim 1.40$ mm C$^{-1}$ d$^{-1}$ corresponds to $\sim 0.05$ mm C$^{-1}$ h$^{-1}$ when $\bar{T}_{10d} > 10$ °C, which agrees
with estimates by Pellicciotti et al. (2005) in isothermal conditions on ice. While establishing a relationship between melt
parameters and $\bar{T}_{10d}$ is novel, note that previous literature has already suggested a seasonal variability in the degree-day





parameter that is conceptually similar to our approach (see Bongio et al., 2016, and references therein). Also, note that $m_r$ should be much smaller than degree-day parameters listed by Hock (2003), because the latter were supposed to account for both the temperature-driven and radiation-driven component of snowmelt. Parameter $m_r$ is set to 0 in case the equation above returned a value lower than 0.

Refreezing is computed when $T_{air} < T_\tau$, using a simple degree-day approach as in Avanzi et al. (2015):

$$R = -c_M m_r (T_{air} - T_\tau).$$ (19)

Compared to Avanzi et al. (2015) or Schaefli et al. (2014), we do not decrease $m_r$ by a reduction factor when computing refreezing. In standard degree-day models that do not separate the temperature and radiation components of melt, that reduction factor conceptually accounts for refreezing melt rate being smaller than snowmelt rate for the same temperature difference, given the usual lack of incoming-shortwave radiation during refreezing-prone periods. This reduction factor is not necessary 235 in S3M, because the contribution of incoming-shortwave radiation is explicitly accounted for in Equation 15 and excluded in Equation 19; in other words, $m_r$ only accounts for turbulent and longwave-radiation factors.

### 2.3.3 Snowpack runoff

While the term *runoff* in catchment hydrology generally denotes *overland flow*, we follow here customary nomenclature in snow hydrology and call snowpack runoff the amount of liquid water discharged by the snowpack (Wever et al., 2014). This 240 flux is parametrized according to a matrix-flow approximation (Colbeck, 1972; Avanzi et al., 2015, 2016):

$$O = \alpha \frac{\rho_W g K_W}{\mu_W},$$ (20)

where $\mu_W$ is water dynamic viscosity, $g$ is acceleration due to gravity, $K_W$ is the intrinsic permeability of water in snow (m$^2$), and $\alpha$ is a time- and unit-conversion parameter (equal to 1000 mm m$^{-1} \times \Delta t$, with $\Delta t$ in seconds). Assuming $\rho_W = 1000$ kg m$^{-3}$, $g = 9.81$ m s$^{-2}$, and $\mu_W = 1.79 \times 10^{-2}$ kg m$^{-1}$ s$^{-1}$ (DeWalle and Rango, 2011), then

$$O = \alpha \alpha' K_W,$$ (21)

with $\alpha' = 5.47 \times 10^5$ m$^{-1}$ s$^{-1}$. We predict $K_W$ following again Colbeck (1972):

$$K_W = K S^{\star 3},$$ (22)

where $K$ is the intrinsic permeability of snow (in m$^2$) and $S^\star$ is the effective saturation degree:

$$S^\star = \frac{Sr - Sr_i}{1 - Sr_i}.$$ (23)



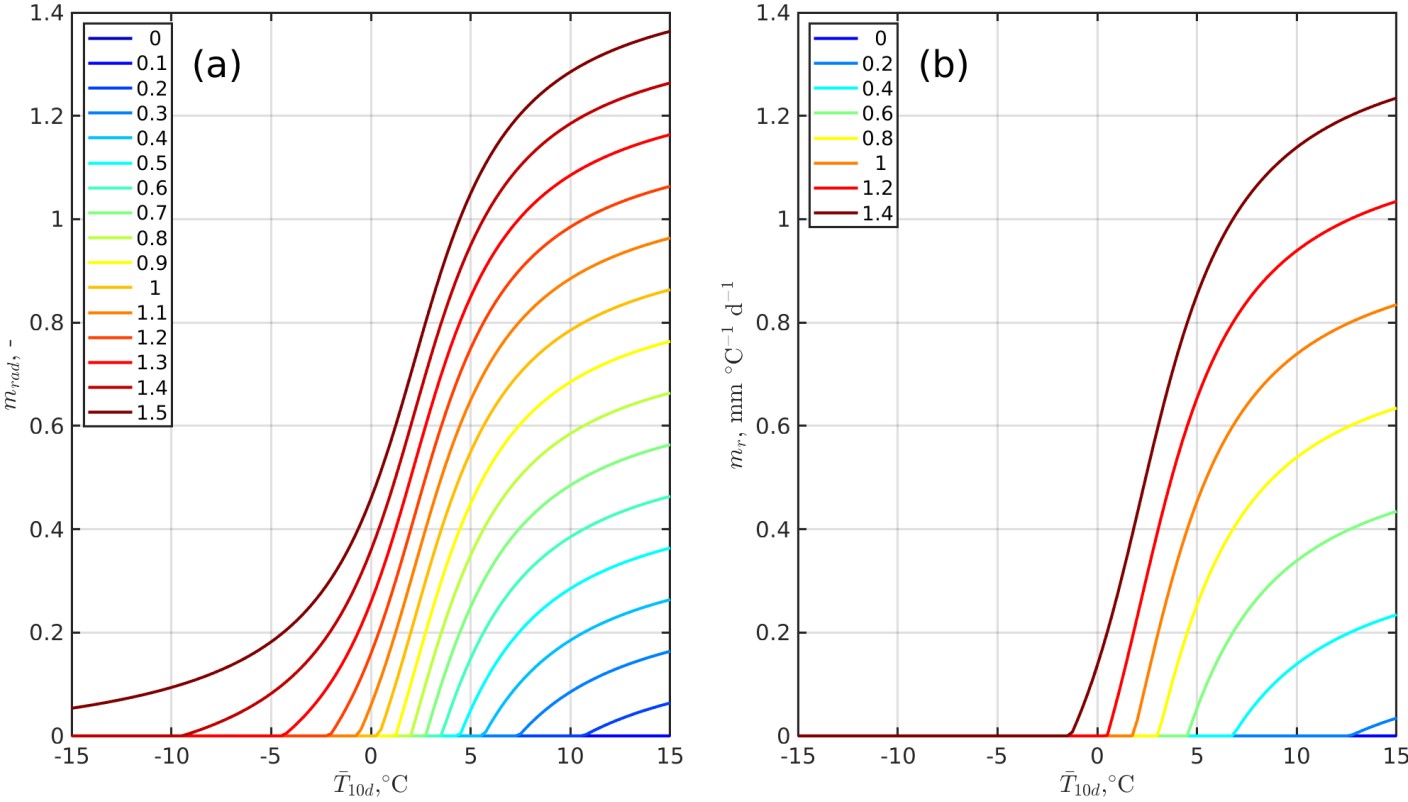

**Figure 2.** Values of the modulating parameter converting shortwave radiation into actual melt ($m_{rad}$, panel a) and the degree-day melt parameter ($m_r$, panel b) as a function of mean air-temperature over the previous 10 days ($\bar{T}_{10d}$) and various values of two tuning parameters ($m'_{rad}$ and $m'_r$, see figure legends for values).

$Sr_i$ is the irreducible saturation degree computed based on Kelleners et al. (2009) as $0.02\rho_D\rho_W^{-1}n^{-1}$, whereas $Sr = h_W n^{-1} h_D^{-1}$ is the saturation degree. Snowpack runoff is set to 0 if $Sr < Sr_i$.

Intrinsic permeability of snow is predicted based on Calonne et al. (2012):

$$K = 3r_e^2 e^{-0.013\rho_D},\tag{24}$$

where $r_e$ is the equivalent sphere radius (m), a conceptual, characteristic length of snow microstructure corresponding "to the radius of a monodisperse collection of spheres having the same specific surface area ($SSA$) as the sample considered" (Calonne et al., 2012). Variable $r_e$ or $SSA$ (m$^2$ kg$^{-1}$) are likely the most objective metrics of snow microstructure to date (Carmagnola et al., 2014), traditional grain size being subjective and cumbersome to measure (Fierz et al., 2009). Still, $SSA$





and $r_e$ are insufficient to fully characterize snow structure, because grain shape also plays an important role in the two-point correlation function (Krol and Löwe, 2016). We relate $r_e$ to $SSA$ by definition:

$$r_e = \frac{3}{SSA\rho_i} \tag{25}$$

and diagnostically estimate $SSA$ following Domine et al. (2007):

$$SSA' = -308.2ln(\rho'_D) - 206, \tag{26}$$

with $SSA'$ being $SSA$ in cm$^2$ g$^{-1}$ and $\rho'_D$ is $\rho_D$ in g cm$^{-3}$. We used Equation 26 to predict SSA because S3M currently does not include a prognostic simulation of snow microstructure. However, Domine et al. (2007) clearly show that density alone is a modest predictor of SSA (R$^2 = 0.43$). Future work will augment snow microphysics to better capture SSA seasonal patterns and how they relate to snow metamorphism (Legagneux et al., 2002; Domine et al., 2013; Carmagnola et al., 2014; Morin et al., 2013), especially in wet conditions (Avanzi et al., 2017; Hirashima et al., 2019).

Following Avanzi et al. (2015), Equation 20 is used as long as $Sr < 0.5$ or $SWE_D > 10$ mm w.e.. This is because Avanzi et al. (2015) report instabilities of Equation 20 for high saturation values or a very shallow snowpack. In situations with $Sr \geq 0.5$ or $SWE_D < 10$ mm, Equation 20 is thus bypassed and $SWE_W$ is directly converted into snowpack runoff.

### 2.3.4 Dry-density equation

Differently from liquid water, bulk dry-snow density is not invariant with time because of three main factors: compaction, new-snow events, and refreezing:

$$\frac{d\rho_D}{dt} = \frac{d\rho_D}{dt}\bigg|_{comp} + \frac{d\rho_D}{dt}\bigg|_{snowf} + \frac{d\rho_D}{dt}\bigg|_{ref}. \tag{27}$$

Regarding compaction, we assume a linear profile for stress with depth and start from the momentum-conservation equation for a representative element at 66% depth:

$$\sigma_v = 0.66\rho_D g h_D = 0.66 SWE_D \rho_W g, \tag{28}$$

with $\sigma_v$ being vertical stress. Equation 28 is then coupled with a viscous rheological equation to obtain, via the definition of the vertical strain rate:

$$\frac{d\rho_D}{dt}\bigg|_{comp} = \rho_D \frac{\sigma_v}{\eta} = \rho_D \frac{0.66 SWE_D \rho_W g}{\eta}, \tag{29}$$





with $\eta$ being viscosity. We finally follow De Michele et al. (2013) and reference therein and define viscosity as an exponential function of dry-snow density and snow temperature:

$$\frac{d\rho_D}{dt}\bigg|_{comp} = 0.66 c_{\Delta t} c_1 \rho_D SWE_D \rho_W e^{0.08T_S - 0.021\rho_D}, \tag{30}$$

or in a more compact form as:

$$\frac{d\rho_D}{dt}\bigg|_{comp} = 0.66 c_{\Delta t} c_1 \rho_D^2 h_D e^{0.08T_S - 0.021\rho_D}. \tag{31}$$

$T_S$ is snow mean temperature (°C), $c_1 = 0.001$ m$^2$ h$^{-1}$ kg$^{-1}$, and $c_{\Delta t}$ is a timestep-adjusting coefficient ($\Delta t \times 3600^{-1}$ s$^{-1}$ h, with $\Delta t$ in seconds).

Because S3M does not solve for the full energy balance, it also does not simulate snow-temperature profiles. If $T_{air} < 0$°C, snow mean temperature is assumed to follow a linear profile between snow-surface temperature and ground surface (assumed equal to $T_{air}$ and 0 °C, respectively, see Ohara and Kavvas, 2006), while it is set to 0°C otherwise.

New events change bulk-snow density proportionally to snowfall depth versus existing-snow depth:

$$\rho_D(t + \Delta t) = \frac{SWE_D + S_f}{\left(\frac{S_f}{\rho_f} + \frac{SWE_D}{\rho_D}\right)}. \tag{32}$$

We handle refreezing with a similar approach to new events, thus implying that refreezing has no impact on snow structure besides the associated volume expansion from liquid water to ice:

$$\rho_D(t + \Delta_t) = \frac{SWE_D + R}{\left(\frac{R}{\rho_i} + \frac{SWE_D}{\rho_D}\right)}. \tag{33}$$

This assumption has been discussed previously, with mixed results (Gallet et al., 2014; Avanzi et al., 2017).

### 2.3.5  Data assimilation

The assimilation framework of S3M is a result of CIMA's operational-forecasting procedures as summarized into the Flood-PROOFS suite (Laiolo et al., 2014; Avanzi et al., 2020a). These procedures – external to S3M – include generating maps of snow depth and satellite-based scene classification (hence, snow-covered area – SCA), as well as processing SWE maps from third parties (e.g., from interpolation of ground manual measurements, see Avanzi et al., 2020a). Given that snow depth and satellite maps are generated with a comparatively high frequency (up to daily), their assimilation in S3M is performed in correspondence to the timestamp to which they refer (this nominal timestamps must be the same for both snow-depth and satellite maps, collectively referred to as Updating maps). SWE maps have various temporal frequencies (usually, weekly),





thus S3M allows the user to specify a temporal window of influence, that is, a period after the official issue date of the SWE map during which the map is assumed to be valid.

  Assimilation of Updating maps also requires a Kernel map and a Quality map. The Kernel ($K$) is generally an output of the geostatistics-based interpolation method used to generate the snow-depth maps and is used to optimally combine observations and model predictions (see below). Instead, the Quality map is used to automatically skip assimilation when values in this

map are below an user-defined threshold. For example, the operational convention in Flood-PROOFS is to forego assimilation during days with large cloud obstruction. Thus, in Flood-PROOFS the Quality map for a given day is computed as the ratio between pixels classified as snow or ground over the total number of pixels (so, in fact, this map reports the same scalar for each pixel). In this way, quality is a measure of the proportion of the satellite map that is covered by clouds. We stress, however, that this quality can be defined based on any user's need, with S3M skipping assimilation below a quality threshold even if the

corresponding snow-depth map was available.

  Prior to assimilation, S3M blends information from the snow depth and the SCA maps based on quality. For example, $K$ is doubled wherever quality is 2.5 times the quality threshold and the satellite indicates ground (ID = 0). Also, snow-depth maps are set to 0 wherever quality is greater than the quality threshold and the satellite indicates ground. Finally, snow-depth maps are set to missing values wherever quality is lower than the quality threshold, modeled SWE is less than 20 mm w.e., and the

satellite indicates clouds, no decision, or no data (ID = 2, 3, and -1, respectively).

  $SWE$ maps, on the other hand, use neither a quality flag nor a spatially distributed Kernel, again owing to how these maps are received and handled by Flood-PROOFS. For assimilating SWE maps, $K$ is thus assumed to be a function of time elapsed since the issue date, with no spatial variability:

$$K = We^{\frac{-(t'-t_{SWE})^2}{0.5\sigma^2}},\tag{34}$$

where $t_{SWE}$ is the official issue time of the SWE map, $t'$ is time relative to this issue date (in days), and $\sigma$ is equal to half of the validity days after SWE-map issue date (user-defined parameter, see the Appendix). On the same date of the issue date, $t' = t_{SWE}$ and so $K = W$, where $W$ is a user-assigned maximum weight of the map to be assimilated. No quality threshold is used for SWE maps given that they are usually the result of reanalysis rather than real-time automatic processing.

  Currently, S3M performs data-assimilation exclusively in the form of SWE. Thus, snow-depth-map assimilation requires a

preliminary step to convert snow depth into SWE via modeled bulk-snow density:

$$SWE_{obs} = \frac{h_{S,obs} \times \rho_{S,S3M}}{\rho_W},\tag{35}$$

  where $h_{S,obs}$ and $\rho_{S,S3M}$ are observed snow depth according to the snow-depth map and simulated bulk-snow density according to S3M, respectively. This step implicitly assumes that snow density is a less relevant source of uncertainty than snow depth in estimating SWE, which is supported by snow-density temporal patterns being consistent from year to year

(Mizukami and Perica, 2008).




Updating and SWE maps are assimilated into S3M using a Newtonian-relaxation approach:

$$SWE_{S3M,post} = SWE_{S3M,prior} + K\left(SWE_{obs} - SWE_{S3M,prior}\right), \tag{36}$$

where $SWE_{S3M,post}$ and $SWE_{S3M,prior}$ are the a-posteriori and a-priori SWE. After assimilating bulk SWE, a few state variables of S3M are modified through factor $U_{SWE}$:

$$U_{SWE} = \frac{SWE_{S3M,post}}{SWE_{S3M,prior}} \tag{37}$$

$$SWE_{D,S3M,post} = U_{SWE} \times SWE_{D,S3M,prior} \tag{38}$$

$$SWE_{W,S3M,post} = U_{SWE} \times SWE_{W,S3M,prior}. \tag{39}$$

This step is needed since total SWE in S3M v5.1 is only a diagnostic variable, and assimilating it does not affect model predictions unless the true prognostic variables are also modified (in this case, $SWE_D$ and $SWE_W$). Factor $U_{SWE}$ assumes
that both the dry and the wet phases are proportionally affected by data assimilation. It is also assumed that dry-snow density does not change during assimilation. Given that dry-density evolution does depend on $SWE_D$, this is a simplification.

S3M also supports assimilating only positive differences in Equation 36, that is, only correcting modeled SWE if observations are larger than simulations. This experimental configuration may help when assimilating observed SWE mainly aims at correcting for precipitation under-catch, while the user would like the ablation season to be unaffected by assimilation. Such an
approach is, e.g., standard in avalanche-forecasting models like SNOWPACK (Lehning et al., 2002). Note, however, that this experimental configuration will likely override the SCA component of the assimilation package, because SCA assimilation in S3M is performed indirectly via setting observed snow depth to zero in areas with observed ground.

The last step in the snow component is to perform a set of sanity checks (e.g., set to zero all state variables where SWE$\rightarrow 0$) and to compute the snow mask, a binary map with the same size of the simulation domain reporting 1 where $SWE > 0.1$
mm, and 0 elsewhere. This mask is an important output of S3M that is sometimes used by hydrologic models to adjust process representation in areas of snow (for example, inhibiting evapotranspiration).

### 2.4 Glacier component

The glacier component of S3M offers three alternative modules: (1) a simple, melt-only approach with no mass balance and no snow-to-ice conversion; this approach is usually the default choice for short-term flood-forecasting-oriented simulations; (2) a
melt-only approach with mass balance but no parametrization of glacier dynamics or snow-to-ice conversion; (3) an approach with a full mass balance, a parametrization of glacier dynamics (the $\Delta h$ parametrization), and snow-to-ice conversion. The last two approaches are the most suited options for long-term, climate-scenario-oriented simulations.





### 2.4.1 G1: Melt-only approach, no mass balance, no glacier dynamics, no snow-to-ice conversion

In the most basic approach, glacier melt takes place on snow-free glacier pixels with a similar parametrization as that of snow
(Equation 15), with only two changes. The first is that glacier albedo $\alpha_G$ is constant (0.4, see Davaze et al., 2018), as opposed
to snow albedo changing with snow age. The second change is that ice-melt rate ($M_G$) on debris-covered glaciers may be
corrected compared to the theoretical melt rate on debris-free glaciers ($M_G^\star$) using a multiplicative reduction factor, $f_{debris}$
(Huss and Fischer, 2016):

$$M_G = (1 - f_{debris})M_G^\star. \tag{40}$$

The correction factor $f_{debris}$ can be estimated with various approaches, for example following Huss and Fischer (2016) who
prescribed it based on the so-called Østrem curve. Accordingly, values of $f_{debris}$ are generally smaller than 0 for very shallow
debris (up to ~5 cm), and between 0 and 1 otherwise (Nicholson and Benn, 2006). S3M expects $f_{debris}$ to be a spatially
distributed, input parameter included in the so-called static-data suite (see Section A). Glacier melt is directly converted into
ice runoff, with no routing (Figure 1). Glacier pixels are defined based on a glacier mask (see Section A).

### 2.5 G2: Melt-only approach, with mass balance, no glacier dynamics, no snow-to-ice conversion

This approach is equivalent to G1 with the only difference that glacier thickness ($h_G$) is dynamically updated every hour based
on glacier melt. To do so, ice melt in mm w.e. according to Equation 15 and optionally Equation 40 is converted to meters of
ice using ice density $\rho_i$. The mass-conservation equation reads:

$$\frac{dIWE}{dt} = -M_G. \tag{41}$$

Wherever $h_G \to 0$ as a result of multi-year melt, ice melt on that pixel is not computed anymore. S3M expects $h_G$ to be a
spatially distributed input (included either in the so-called restart data or in the static data, see Section A).

### 2.6 G3: Full mass-balance approach with glacier dynamics and snow-to-ice conversion

This approach – ideal for multi-year simulations – includes snow-to-ice conversion and a specific mass-redistribution approach
called the $\Delta$h parametrization (Huss et al., 2010), which allows one to implicitly account for glacier-movement effects without
implementing a full ice-flow model. Because the $\Delta$h parametrization is better suited for multi-year time scales rather than day-
to-day thickness changes, module G3 requires the user to define the month of water-year start, so that S3M will accumulate
glacier melt for each pixel throughout the water year and update $h_G$ at the beginning of each new water year. Regardless of
this accumulation procedure, ice melt is still outputed every time step, so that seasonality in runoff-generation processes is
preserved. In other words, this accumulation procedure only regards changes in glacier thickness and not ice-runoff generation.
This is important in case S3M was coupled with a hydrologic model.



Snow-to-ice conversion is performed by simply prescribing that, on pixels with $h_G > 0$, any residual SWE at the end of each water year is added to $h_G$. Consequently, SWE as well as all snow-related bulk state variables are reset to 0. This pragmatic approach is not performed in areas where SWE$> 0$ and $h_G = 0$ at the end of the season; in such conditions, the snowpack is maintained through the start of the new water year.

As for the $\Delta h$ parametrization, this is presented in Huss et al. (2010) and further discussed for hydrologic models by Seibert et al. (2018), so we limit ourselves to a short overview here. This approach starts from the empirical intuition that glacier-thickness changes as a result of both the mass balance and glacier flow have recurring patterns throughout seasons of persistently negative mass balances (Huss et al., 2010). By parametrizing these recurring patterns, the $\Delta h$ parametrization allows one to simulate the effect of movement in addition to the mass balance without a complex ice-flow model. These patterns

are derived through differentiating two digital surface models (Huss et al., 2010) and then fitting a glacier-specific power law like

$$\Delta h = (h_r + a)^\gamma + b(h_r + a) + c, \tag{42}$$

where $a, \gamma, b, c$ are calibration parameters, $\Delta h$ is the change in surface elevation (normalized by the maximum decrease across all glacier pixels), and $h_r$ is normalized glacier elevation defined as

$$h_r = \frac{h_{max} - h}{h_{max} - h_{min}}, \tag{43}$$

with $h_{max}$ and $h_{min}$ being the maximum and minimum elevations of that glacier at the beginning of the water year and $h$ being glacier elevation at a given pixel, respectively. Note here that $h$, $h_{max}$, and $h_{min}$ are *elevations*, not thicknesses like $h_G$.

In practice, applying the $\Delta h$ parametrization requires (1) assigning an ID to each glacier for which the user would like to use the $\Delta h$ parametrization (S3M expects this ID to be a positive integer); (2) mapping these glaciers ID on the simulation

raster, so that S3M will be able to identify all pixels of the simulation domain belonging to a given glacier; (3) a-priori deriving Equation 42 for each glacier of interest and passing it to S3M as a pivot table, where $\Delta h$ is sampled for a number of discretized $h_r$. S3M will then assign a $\Delta h$ for each pixel of a given glacier using a nearest-neighbor interpolation of this pivot table. Both the glacier-ID map and the pivot table are part of the so-called static-data suite in input to S3M (Section A).

Once these preliminary steps are performed, S3M computes ice melt as in module G2, but $h_G$ is not dynamically updated.

Instead, ice melt for each pixel is accumulated to yield $b_a$, the cumulative mass balance in mm w.e. At the end of the water year, this information is used to compute factor $f_s$, which is employed to *scale* Equation 42 (the $\Delta h$ profile) and so derive the actual change in glacier thickness for each glacier pixel:

$$f_s = \frac{\sum_{i=1}^{i=N_G} \left[ b_{a,i} \times A_i \times A_{tot}^{-1} \right]}{\sum_{i=1}^{i=N_G} \left[ \Delta h_i \times A_i \times A_{tot}^{-1} \right]} \tag{44}$$

where $i$ denotes the $i$-th pixel of a given glacier, $N_G$ is the total number of pixels of that glacier, $A_i$ is the area of each pixel,

and $A_{tot}$ is the total area of the glacier. Once $h_{G,i} \to 0$, that pixel is excluded from all computations and $h_{max}$ and $h_{min}$ as well all other variables are updated accordingly.





The general mass-conservation equation for glacier pixel $i$ using G3 therefore reads:

$$IWE_{i,WY+1} = IWE_{i,WY} + SWE_{i,r} - f_s \Delta h_i, \tag{45}$$

with $SWE_{i,r}$ the residual snow water equivalent on that pixel at the end of the water year, while $IWE_{i,WY+1}$ and
$IWE_{i,WY}$ are ice-water equivalent for piel $i$ during the current and the previous water year ($WY$), respectively.

The implementation of the $\Delta h$ parametrization in S3M provides a number of modeling degrees of freedom. First, S3M assumes a non-dynamic mass balance for all pixels that are part of the model glacier mask, but have no glacier ID assigned; the user can explicitly choose this approach also for glaciers having a glacier ID, by setting all entries of the corresponding pivot table to -9999. This option is useful either where fitting a $\Delta h$ parametrization is cumbersome (e.g., spatially incoherent
glaciers) or for glacial remnants that are not moving anymore (e.g., glacierets). Second, the modeler can use different $\Delta h$ parametrizations for various parts of the same large glacier, by simply assigning different glacier IDs to these parts and providing specific pivot-table entries for each ID. Also note that the $\Delta h$ parametrization is bypassed whenever one glacier occupies only one pixel.

## 3   Case study: Aosta valley, NW Italian Alps

S3M is open software, including algorithms to prepare input data and set up computational environments and libraries. Links to all code are reported in the User Manual (see Section A), with a general reference being CIMA Foundation's Hydrology and Hydraulics repository at https://github.com/c-hydro (Github organization).

This Section presents an application of S3M for an inner-Alpine valley located in north-western Italy (Aosta valley, Figure 3). This area has steep elevation gradients, with the main valley at elevations on the order of 300-400 m ASL and peaks as
high as 4800 m ASL (Mont Blanc) or 4478 m ASL (Matterhorn). About 4% of Aosta valley is covered by glaciers (134 km$^2$), some of which are characterized by thick debris and develop for several kilometers (e.g., the Miage glacier in the Mont-Blanc massif). With its cryosphere-dominated water supply and complex precipitation-topography interactions leading to marked rain shadows (Avanzi et al., 2020a), Aosta valley is a formidable test bed for S3M.

S3M has been operational in Aosta valley since the early 2000s, as a component of a flood-forecasting chain called Flood-
PROOFS (see Laiolo et al., 2014; Avanzi et al., 2020a, and Section 2). This chain includes algorithms to spatialize and downscale weather-input data, automatically generate daily maps of snow depth and use MODIS snow-covered area, and process independently derived weekly maps of SWE. Together with runs of S3M in assimilation mode at ~240 m spatial resolution, these tools inform real-time forecasts of streamflow at relevant closure sections obtained using the Continuum model (Silvestro et al., 2013). Details about spatialization techniques and hydrologic modeling in Flood-PROOFS are available elsewhere (Boni
et al., 2010; Laiolo et al., 2014; Avanzi et al., 2020a) and are not discussed here for brevity. In the present paper, we instead leverage our application in Aosta valley to provide guidelines on how to calibrate S3M in a real-world case study (Section 3.1) and how to validate and interpret model results for the snow (Sections 3.2 to 3.4) and the glacier component (Section 3.5).

**Figure 3.** Aosta valley in north-western Italy. Panel (a): topography and glaciology of this region, with location of all snow-evaluation data used in this paper; panels (b) and (c): zoom on two intensive measurement regions, the Rutor glacier and the headwaters of Valpelline catchment, respectively. "HS sensors" are continuous-time snow-depth ultrasonic sensors, "Torgnon" is an intensive study plot with a variety of snow and weather datasets (see Section 3.3 and Terzago et al., 2019), "Aval. probes" denotes locations of snow-depth measurements for avalanche-forecasting purposes, "Weekly samples" denotes locations of ∼weekly snow-depth measurements collected mainly for water-supply forecasting, while "Peak-HS courses" are snow-depth measurements collected along transects of several kilometers for hydropower-forecasting purposes (see Avanzi et al., 2020a, for more details).

### 3.1   Calibrating S3M

In principle, a fairly large amount of S3M parameters can be calibrated, or at least fine-tuned based on expert knowledge. These
parameters include the radiation and temperature snowmelt parameters $m'_{rad}$ and $m_r$, the threshold temperature for inhibiting



snowmelt ($T_\tau$), the maximum weight of SWE-assimilation maps ($W$), as well as a number of climatological thresholds used to constrain model predictions (e.g., maximum and minimum snow density, or the snow-quality threshold to enable data assimilation, see Section A and Table A1). In practice, our experience suggests that the two melt parameters, $m'_{rad}$ and $m_r$, are the prime calibration parameters of this model.

Because S3M is employed in assimilation mode in Aosta valley, and this mode includes MODIS snow-covered area, our calibration rationale focused on maximizing fit for point predictions rather than for spatial patterns. Still, calibration was performed considering open-loop simulations to avoid model performance to be spuriously driven by assimilation rather than parameter values. Calibration data comprised spatially distributed continuous-time measurements of 53 snow-depth sensors and temporally discontinuous manual measurements of snow depth collected across the region for water-supply or avalanche

forecasting (see Avanzi et al., 2020a, and Figure 3 for a data inventory). The calibration period covered water years 2010 through 2019, where the bulk of evaluation data was concentrated; the water year is a period between September 1 to August 31 and is indicated with the calendar year when it ends.

Our calibration protocol was based on performing multi-year simulations of S3M for a range of values of $m'_{rad}$ and $m_r$: [0.8, 2] and [0.5, 1.5] mm °C$^{-1}$ day$^{-1}$, respectively. These ranges were chosen based on preliminary tuning and, importantly,

the notions that $m'_{rad} \sim 1.1$ implies a 100% efficiency of transmitted shortwave radiation in generating melt under likely isothermal conditions ($\bar{T}_{10d} \to 10°C$, see Figure 2(a) and Section 2), while $m_r \sim 1.4$ tallies with previous work by Pellicciotti et al. (2005). The parameter space was explored for increments of 0.025 of $m'_{rad}$ and 0.05 mm °C$^{-1}$ day$^{-1}$ of $m_r$, and we first calibrated an optimal value of $m'_{rad}$ and then of $m_r$. This meant running 29 decade-long simulations for all tentative values of $m'_{rad}$, finding the optimal one, and then re-running 25 simulations tuning $m_r$. This sequential calibration approach is only

one out of several possible calibration approaches, including strategically exploring the parameter space. Here, we chose a sequential approach for this illustrative case study mainly for computational-resource constraints.

As objective metric, we minimized $Obj(\Theta) = 0.5 \times (1 - KGE(\Theta)) + 0.5 \times (RMSE(\Theta))/\overline{RMSE(\Theta)}$, with $\Theta$ being the generic parameter value, KGE being the Kling-Gupta Efficiency metric as defined in Kling et al. (2012), RMSE being the Root Mean Square Error, and $\overline{RMSE(\Theta)}$ being the mean RMSE across all parameter values. This objective metric was chosen

to combine the features of KGE (and in particular its focus on correlation, bias, and ratio of coefficients of variation) with a specific weight for large errors (RMSE). The RMSE was normalized by its mean across all parameter values to make it adimensional and so comparable to KGE. For each parameter value $\Theta$, $Obj$ was computed between observed snow depth (Figure 3) and simulated $h_S$, both merging all data from all years in one sample and separately for each water year (all-years and yearly values, respectively).

Median yearly values across all water years show a minimum for $m'_{rad} = 1.125$ and then $m_r = 1.10$ mm °C$^{-1}$ day$^{-1}$, in line with expectations (Figure 4 and Section 2). Objective-metric values showed remarkable variability across water years (see the quartile range in Figure 4), owing to significant variability in the original sample of snow-depth values between warm and cold years (Avanzi et al., 2020a). Thus, multi-year calibration periods are recommended, although the range of variability in median and all-year $Obj$ in Figure 4 suggests that the sensitivity of S3M to $m'_{rad}$ and $m_r$ is surprisingly low, especially if one considers

that calibration-based snow models are prone to large drops in performance outside the calibration sample (Hock, 2003; Avanzi



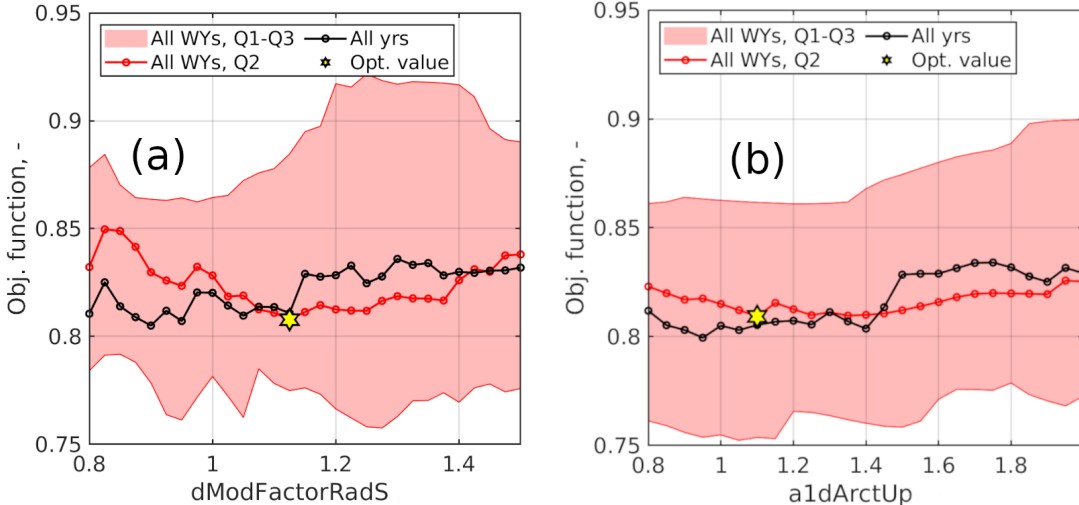

**Figure 4.** Calibration objective metrics as a function of parameter values, with dModFactorRadS being $m'_{rad}$ and a1dArctUp being $m_r$ (see the Appendix for details on model's notation in the source code). Parameter $m'_{rad}$ is adimensional and $m_r$ is in mm °C$^{-1}$ day$^{-1}$. For each parameter value $\Theta$, the objective metric was computed using both all data from all years in one sample and separately for each water year (all-years and yearly values, respectively). Q1, Q2, and Q3 are the first, second, and third quartiles of objective metrics across all water years. WY is water year.

et al., 2016). We interpret this to be because of the hybrid physics-based and temperature-index approach employed in S3M to predict snowmelt (Equation 15), which appears to better constrain parameter values than a purely temperature-index approach.

### 3.2 Evaluation: point snow depth

Figure 5 show simulated vs. observed snow depth (symbol HS per guidelines by Fierz et al., 2009) for the 52 snow-depth
sensors in Figure 3, the snow-depth sensor at the Torgnon study plot, as well as for all manual snow-depth measurements taken for avalanche, hydropower, and water-supply forecasting (panels a to e, respectively). The evaluation period in Figure 5 and all the remaining sections of this paper is water years from 2004 to 2009 and 2020, that is, all years before and after the calibration period in Section 3.1. Simulations in this and following Sections were carried out in assimilation mode, as this is the approach we generally use in real-time forecasting and so results are more representative of real-world model performance. More details
on this assimilation procedure are reported in Avanzi et al. (2020a); note that snow-depth assimilation maps are only generated between August and April in an effort to unaffect the simulation of the depletion phase of the seasonal snowpack.

Snow-depth-sensor and weekly-snow-sample measurements in Figure 5(a) and (e) were indirectly assimilated in S3M as they are involved in the computation of the so-called Updating and weekly-SWE maps (see Section 2), meaning their performance statistics are only reported for reference. Non-assimilated data at Torgnon (Figure 5(b)) and from avalanche probes (Figure

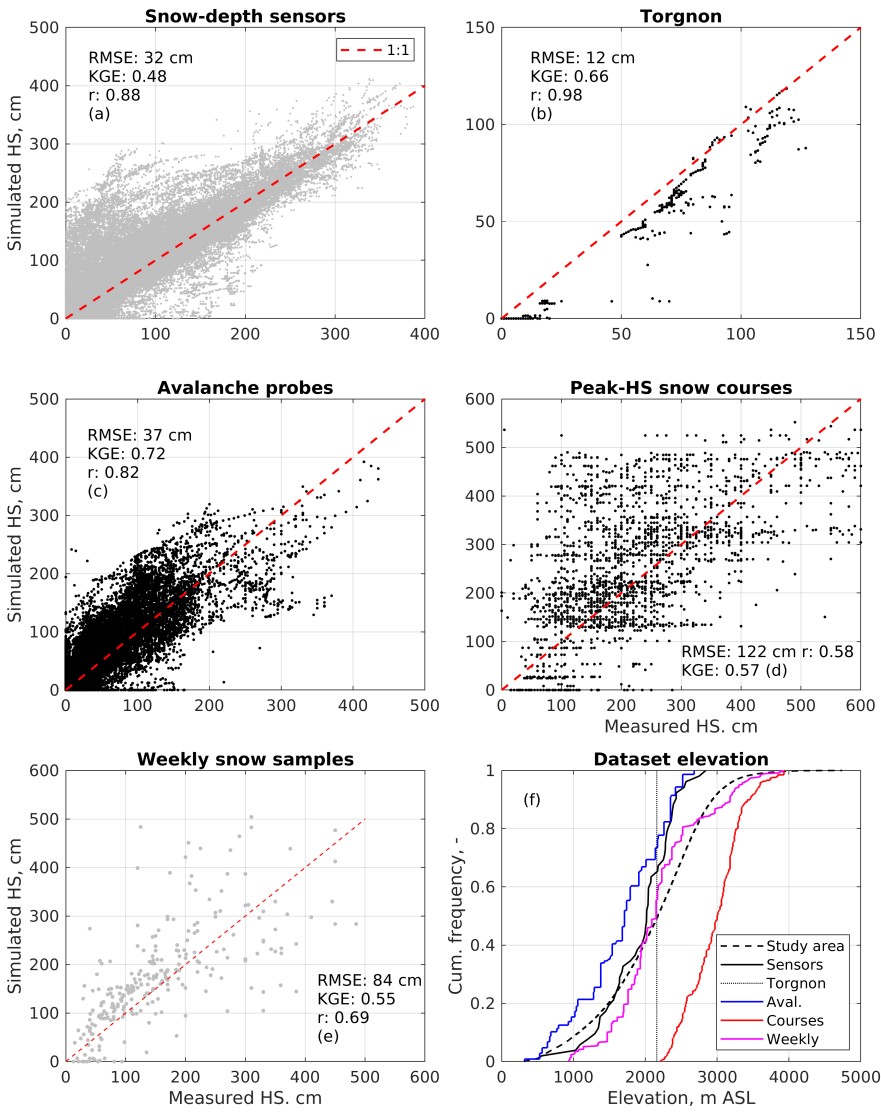

**Figure 5.** Performance of S3M in simulating point snow depth, as measured by continuous-time snow-depth sensors (a), the snow-depth sensor at the intensive study plot of Torgnon, and manual snow-depth measurements taken for avalanche, hydropower, and water-supply forecasting (c to e). Because simulations were carried out in assimilation mode, and assimilated maps indirectly involved snow-depth sensors and weekly snow samples, performances for these datasets are reported only for reference. Panel (f) shows the elevation distribution of each dataset and of Aosta valley for context. RMSE is Root Mean Square Error, KGE is the Kling-Gupta Efficiency (Kling et al., 2012), and r is Pearson's correlation coefficient.

5(c)) maintain comparatively high values of RMSE and KGE (12-37 cm and 0.66-0.72, respectively), with no evident tendency for systematic under- or overestimations. Open-loop results were not reported here for brevity, but generally show comparable





performance for these datasets. Thus, we conclude that our calibration strategy not only showed little sensitivity of the model to $m'_{rad}$ and $m_r$ (Figure 4), but also led to a robust performance across nearly twenty water years and for areas that were not included in the calibration pool.

Peak-snow-depth courses show a significantly lower performance, in particular in terms of a larger dispersion around the 1:1 line, a larger RMSE (122 cm), and a lower KGE (0.57, (Figure 5(d)). This is because this course dataset comprises snow-depth measurements at significantly higher elevations than any other considered dataset (see elevation distribution in Figure 5(f)). At those elevations, both interactions between the snowpack and topography complicates snow distribution compared to low-elevation areas, and density of input-data stations is much lower (Avanzi et al., 2020a). Still, the fact that S3M does not show

any obvious over- or underestimation for those elevations is encouraging for our scopes, also considering the comparatively coarse resolution of our implementation in Aosta valley (∼240 m).

### 3.3    Evaluation: the Torgnon study plot

S3M satisfactorily reproduces the timing of peak accumulation and so the onset of the snowmelt season at the Torgnon study plot, both in terms of snow depth ($h_S$) and SWE (Figure 6). Discrepancies between observed and simulated snow depth increase

in spring, mainly because spatial heterogeneity in snow depth increases during the snowmelt season (Grünewald et al., 2010) and this challenges the comparison between a point measurement of snow depth and our 240-m snow model (this could be improved by increasing model resolution). Overall, the model correctly captures snowmelt rate and peak SWE for most water years, which are the primary variable of interest in operational snow hydrology.

S3M also reproduces the magnitude of bulk-snow density and its increase with time for all water years (Figure 6), in

agreement with previous models implementing a similar parametrization of snow settling (De Michele et al., 2013; Avanzi et al., 2016) and despite its one-layer approach. Values of bulk and dry snow density are very close to each other during the accumulation season, while the latter diverges from the former during the snowmelt season. This is due to an increase in mass for the wet component of the snowpack during spring, as confirmed in terms of bulk liquid-water content (Figure 6). The seasonal range of variability for modeled bulk liquid-water content and its peak around 5 vol% during the snowmelt season

agree with measurements by Techel and Pielmeier (2011); Heilig et al. (2015); Avanzi et al. (2017) and the international classification by Fierz et al. (2009).

The Torgnon study plot also measures incoming and reflected short-wave radiation, which allowed a comparison in terms of measured and simulated albedo (Figure 6). During the accumulation season, measured albedo is generally higher than simulated albedo; in particular, both measured and simulated albedo show maximum values around 0.95, but only the latter

decreases well below 0.8-0.7 between snowfall events. Simulations by SNOWPACK at the same study plot (Terzago et al., 2019, not reported for brevity) qualitatively showed higher values than S3M, evidence that only relying on time as a predictor of albedo may yield frequent underestimations compared to a model that considers a broader spectrum of albedo predictors like SNOWPACK. On the other hand, S3M well captures the measured decline in albedo during the snowmelt season, which, again, is important to capture the timing and intensity of seasonal melt.



**Figure 6.** Comparison between measurements and modeling outputs at the Torgnon study plot. First, second, third, and fourth columns are snow depth (HS), SWE, bulk-snow density, and albedo, respectively. The third and fourth columns also report dry-snow density and bulk volumetric liquid-water content, respectively. Each row is one water year (2013 to 2020).





## 3.4 Evaluation: snow distribution


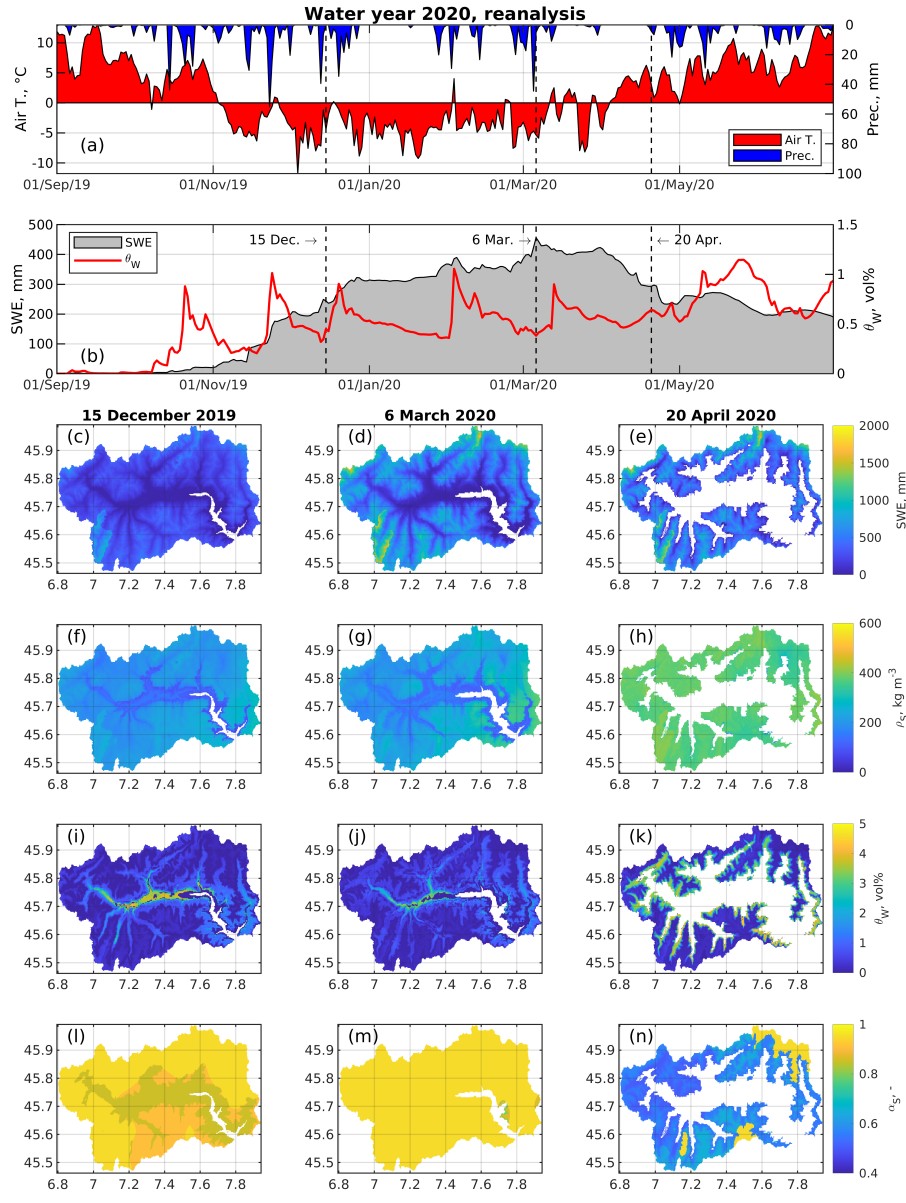

**Figure 7.** Simulated reanalysis of the 2020 snow season: spatially average air temperature and total precipitation (a), spatially average SWE and bulk liquid-water content $\theta_W$ (b), and distribution of SWE (c-e), bulk snow density $\rho_S$ (f-h), bulk liquid water content (i-k), and albedo $\alpha_S$ (l-n) for three example dates: 15 December 2019 (first column), 6 March 2020 (middle column), and 20 April 2020 (third column).

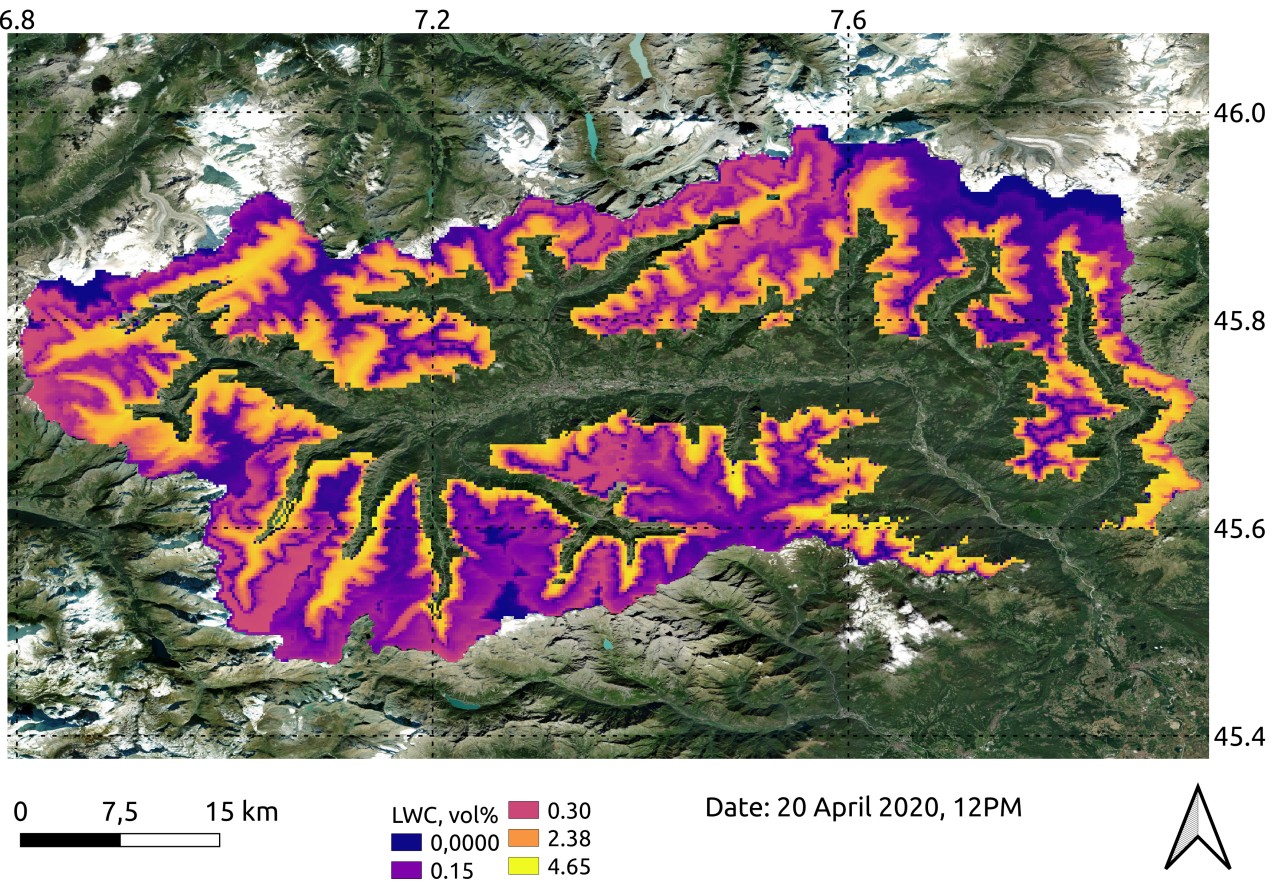

**Figure 8.** Map of bulk volumetric liquid-water content for Aosta valley (LWC, same as $\theta_W$), 20 April 2020 at 12PM. Background layer is from the ESRI Satellite theme.

Figure 7 shows a simulated reanalysis of the 2019-20 snow season, which well exemplifies the information and level of details provided by S3M to forecasters. The 2020 snow season started by the end of October 2019 (Figure 7(a)), with largely uninterrupted precipitation events between November and December 2019 leading to nearly 75% of spatially average SWE across Aosta valley being accumulated before January 2020 (Figure 7(b)). January and February were relatively dry months, with only one warm storm in mid-February and then a cold one between February and early March 2020. The snowmelt season started in April, even though ∼40% of spatially averaged SWE at high elevations persisted by the end of May 2020. The season was characterized by an alternation between cold and warm spells, which led to frequent melt-freeze cycles in the spatially averaged snowpack (Figure 7(b)).

The spatial distribution of SWE confirmed an increase in snow accumulation between 15 December (Figure 7(c)) and 6 March (Figure 7(d)), with the expected positive gradient with elevation. Simulations for 20 April 2020 showed a typical snapshot of the snowmelt season, with largely depleted snowpack at low and medium elevations and SWE still on the order of





1000 - 1500 mm at elevations above 3000 m ASL. Depletion was spatially more extensive on south-facing slopes compared to north-facing slopes, due to topography-radiation interactions.

Bulk-snow density was spatially fairly homogeneous, especially at the beginning of the snow season (Figure 7(f)). With time,
some differences emerged, with snow density increasing faster in areas with both larger SWE and likely warmer temperatures (Figure 7(g) and (h)). Both the magnitude of snow-density values in Figure 7 and the fact that this variable was spatially more homogeneous than SWE tallies with previous works (López Moreno et al., 2013).

Maps of bulk-liquid water content were largely influenced by local climate, with a general rise in wetness with decreasing elevation that closely followed local topographic contours and aspect (Figure 7(i) to (k) and see Figure 8). Overall, bulk liquid-
water content around the snow line was larger in April than in December or March, which again tallies with expected seasonal trends in wetness (Techel and Pielmeier, 2011). While liquid water in snow has been investigated for a long time (Colbeck, 1971), recent work on wet snow provides renovated opportunities for considering bulk liquid-water content from an operational standpoint, whether to predict the onset of snowpack runoff (Wever et al., 2014) or wet-snow avalanches (Wever et al., 2016). Recent evidence that wet-snow conditions have increased in frequency and have extended well into the winter season due to a
warming trend (Pielmeier et al., 2013) further justifies interest in spatially explicit, raster-like predictions of wet-snow patterns like those in Figure 8. S3M is among the first parsimonious snow models to provide such information.

Maps of albedo showed the expected homogeneity during the accumulation season, owing to frequent snowfall events between November and March (Figure 7(l) and (m)). On the other hand, albedo in spring was much lower and spatially more diverse than in winter, with residual high values across the highest peaks of Aosta valley and values well below 0.6-0.7 in areas
covered by older and wetter snow.

## 3.5 Evaluation: glacier evolution

The dataset considered while evaluating S3M in Aosta valley comprised 94 ablation-stake measurements collected between 2009 and 2015 across the Rutor, Timorion, and Petit Grapillon glaciers (Figure 9). These are all high-elevation, debris free glaciers of various size (7.91 km$^2$, 0.48 km$^2$, and 0.18 km$^2$, respectively – 2012 data), thus providing a representative sample
to test the accuracy of S3M in capturing glacier ablation. Simulations using the G3 glacier module (that is, including the $\Delta h$ parametrization) returned solid results in this regard, with a correlation between simulated and observed change in thickness of 0.6 (Figure 9(d)). The correspondence between simulated and observed change in thickness across the total range of variability in measurements was visually higher for the Rutor glaciers than for the Timorion and the Petit Grapillon, which we interpret because of the large size of the first compared to model resolution (240 m). Also, the number of available samples for the Rutor
is significantly larger than for the Timorion and the Petit Grapillon, meaning the performance observed for the former is likely more representative of S3M predictive skills.

The evolution of glacier thickness for the Rutor glacier shows expected spatial patterns, with minor ablation at elevations above 3000 m ASL and progressively more intense melt close to the terminus below 2750 m ASL (Figure 10). Annual changes show significant interannual variability, with somewhat more intense melt as the evaluation period progresses; in any case, the





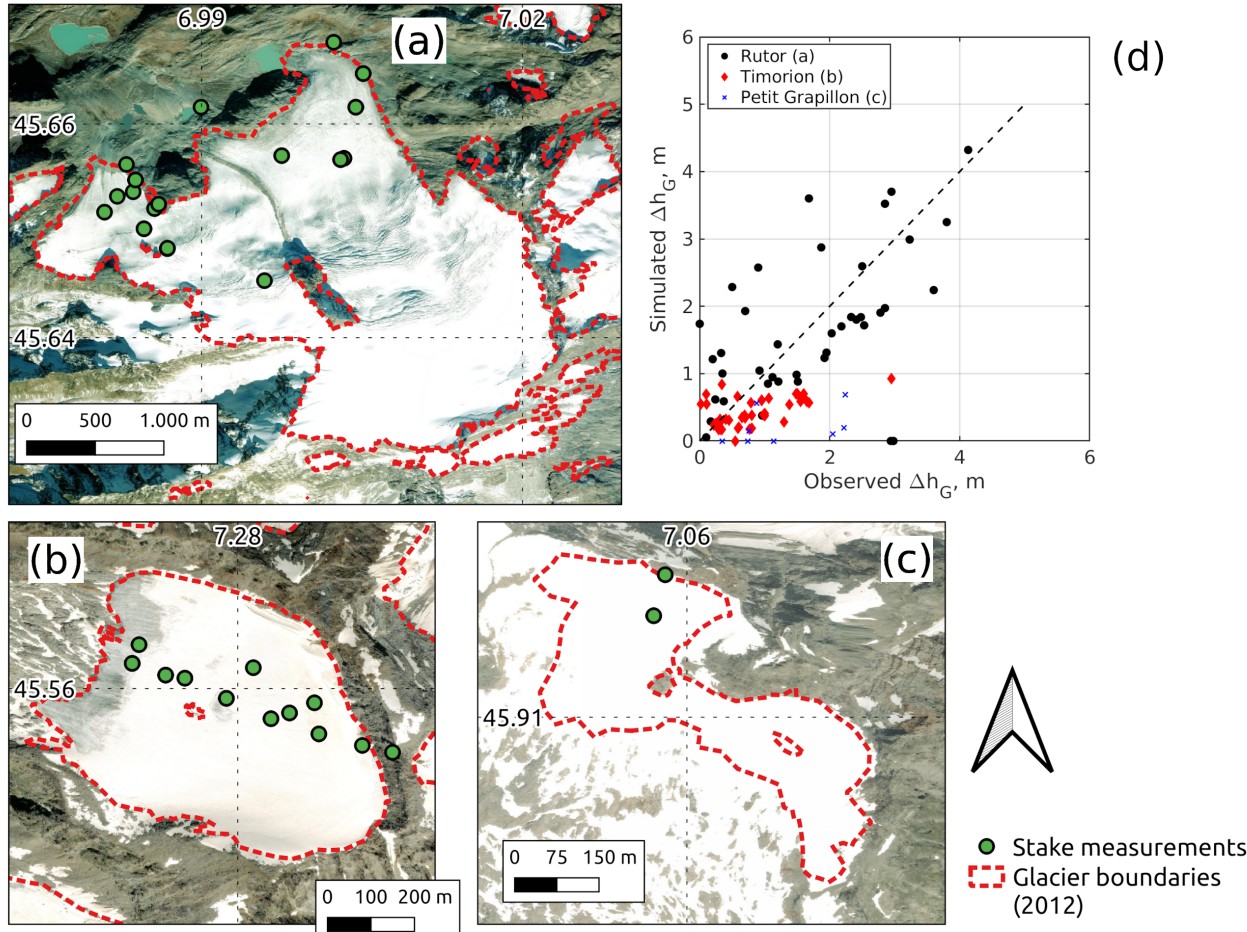

**Figure 9.** Spatial distribution of glacier-ablation measurements across the Rutor (a), the Timorion (b), and the Petit Grapillon (c) glaciers in Aosta valley. Panel (d) shows a comparison between measured and simulated ablation (positive values mean a decrease in local ice thickness). Background layer is from the ESRI Satellite theme.

spatial pattern is preserved as hypothesized by the Δh parametrization. At elevations above 3200 m ASL, glacier thickness increased owing to snow-to-ice conversion.

     Figure 11 shows similar spatial patterns for one of the most complex glaciers in the Alps, the Miage glacier, a 10.8-km$^2$ (2012), 10-km+ long valley glacier covering a ∼ 2000-m elevation range of the Mont Blanc massif. Like many other valley glaciers across the southern Alps (Diolaiuti et al., 2003), vast portions of the Miage tongue are covered by debris, which

has been shown to lead to below-debris melt being insensitive to variations in atmospheric temperature (Brock et al., 2010). Albeit hard to validate due to a lack of measurements, our implementation of the Miage glacier qualitatively captured this disconnection between intense melt across medium-elevation areas with little to none known debris and low melt rates in areas close to the glacier terminus that are well known to be covered by thick debris (Figure 11). Thanks to supporting a spatially



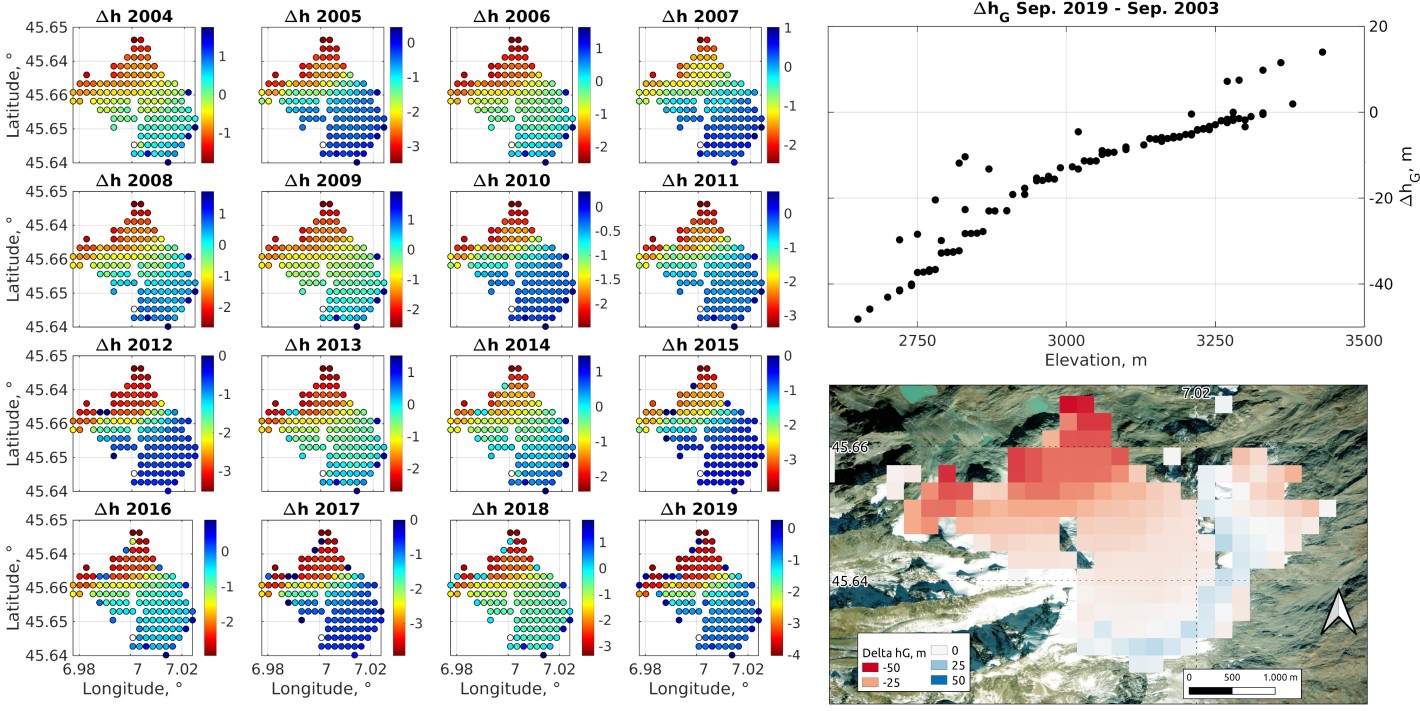

**Figure 10.** Rutor glacier: spatial patterns of annual change in glacier thickness (left), cumulative change in glacier thickness between September 2003 and 2019 as a function of pixel elevation (right, top), and spatial distribution of this cumulative change (right, bottom). Background layer is from the ESRI Satellite theme.

explicit debris-driven melt factor (Equation 40), S3M yielded estimates of thickness change that were much more spatially diverse on the Miage than on the Rutor glacier (and so much less correlated with elevation, compare Figure 11 with Figure 10). At elevations above ∼3000 m ASL, debris is residual if non-existent, and so change in thickness and elevation maintained the same high correlation observed on the Rutor glacier in Figure 10.

## 4 Applicability and future developments

As a hydrology-oriented cryospheric model, S3M delivers timely and computationally efficient predictions of the most significant features of the cryosphere water budget, while still aiming at a comparatively high standard in physical realism. Understanding this trade-off and its implications is important to determine model applicability and adequacy in a given context.

The structure and state variables of the model are all geared toward providing decision-relevant information for water-supply forecasting, which remains the prime area of application tested by the authors (Laiolo et al., 2014). In this context, typical questions that S3M contributes to answer are: how much snow-water resources are currently accumulated across the landscape? What headwater regions are currently releasing meltwater, and what are still accumulating snowpack? When a



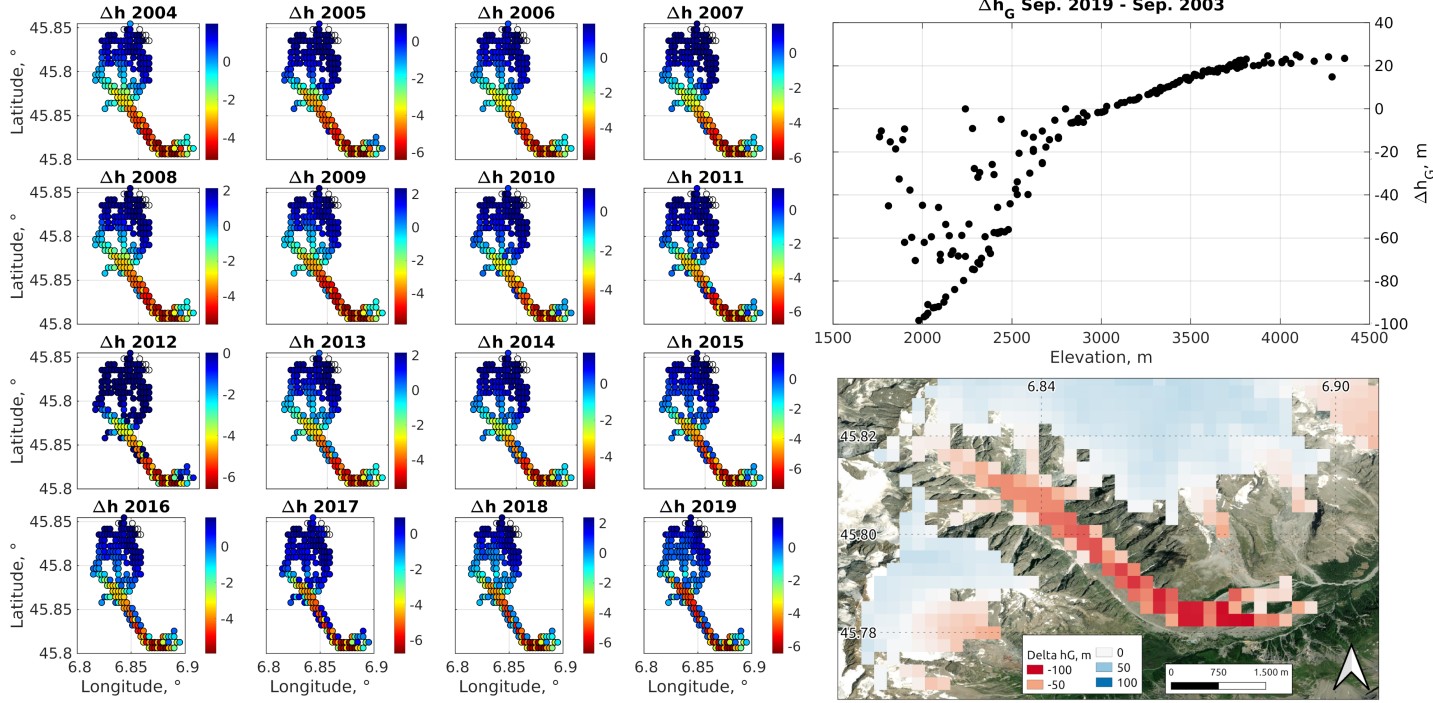

**Figure 11.** Miage glacier: spatial patterns of annual change in glacier thickness (left), cumulative change in glacier thickness between September 2003 and 2019 as a function of pixel elevation (right, top), and spatial distribution of this cumulative change (right, bottom). Background layer is from the ESRI Satellite theme.

certain percentage of the seasonal freshet is expected to reach a given closure section? Are glaciers currently contributing runoff, and if so how much is their relative contribution? Decisions that are currently being informed by S3M thus range from flood-forecasting early warning to hydropower planning. S3M also provides some support to avalanche forecasting, but this remains an unexplored area of application for which the microstructural detail in this model is largely insufficient.

Recently, we also started using S3M to produce future scenarios of water resources in mountain regions. Two attractive features of S3M in this regard are the comparatively limited computational times of this model and the inclusion of both snow and glacier mass balances. Regarding the former, computational time for one water year worth of simulations in Aosta valley runs in the hours for an ordinary laptop with solid-state storage (240 m resolution, ~3290 km$^2$). Storage type is a factor here because S3M includes access to, and creation of, input and output NetCDF files (see the Appendix), and the frequency and

size of output files in particular play a key role in determining computational time. Regarding snow and glacier mass balances, models providing even medium-level physical realism of both these features are still rare, not to mention the scarcity of open-source suites reconnecting them with the hydrologic cycle and so with streamflow predictions (Bongio et al., 2016; Li et al., 2015). Thanks to its integration with the Continuum hydrologic model (also open source, see Silvestro et al., 2013), S3M can deliver such mass-conserving and spatially consistent predictions of the entire mountain water budget.





S3M is currently being actively maintained and further developed, with four main areas of planned future work. The first is the inclusion of wind effects, both as an additional component of the energy balance and as a driver of snow redistribution (wind drift). Explicitly including wind in the energy budget is particularly urgent given that turbulent fluxes are a key contributor to melt during flood-generating rain-on-snow events (Marks et al., 1998; Würzer et al., 2016). As for wind drift, advances in this context are warranted not only in terms of relocation of blowing snow in the form of suspension, saltation, and creep

(DeWalle and Rango, 2011), but also (and likely more importantly) in terms of the associated sublimation. Progress in this regard has been hampered by a lack of detailed measurements of wind across the complex terrain of mountain headwaters, but the publication of recent datasets in this regard may favor future work on this topic (Guyomarc'h et al., 2019).

    A second area of planned future work regards the inclusion of vegetation effects on the snowpack. The science of canopy-snow interactions has identified four mechanisms through which vegetation can alter snowpack evolution compared to open

areas: precipitation interception and throughfall, shortwave radiation shadowing, longwave-radiation enhancement, and wind shielding (Rutter et al., 2009). While the importance of each of these mechanisms for the fate of a seasonal snowpack dramatically changes with local climate (Lundquist et al., 2013), scientific consensus is that canopy may reduce peak SWE by more than 50% and lead to perturbations in the melt-out date on the order of weeks (Rutter et al., 2009), depending on canopy or snow-fractional cover. Helbig et al. (2020) have recently proposed a parsimonious parametrization for canopy effects in large

scale models, thus providing a solid starting point to include these processes in S3M.

    The third direction of future development is liquid-water transport in snow, a rarely parametrized but important connection between surface melt and snowpack runoff. Water infiltration through snow manifests itself as both spatially homogeneous matrix flow and spatially heterogeneous preferential flow (Katsushima et al., 2013), with transition between these two regimes being driven by wet-snow metamorphism and snow properties like density and grain size (Avanzi et al., 2017; Hirashima et al.,

2019). While capturing such micro-scale mechanisms is likely beyond the scope of a large-scale, distributed model, including some forms of preferential flow in S3M will likely enhance its performance in terms of timing and peak of early-season snowmelt events or rain on snow (Wever et al., 2014; Würzer et al., 2017). A way forward in this regard may be the simple parametrization originally proposed by Katsushima et al. (2009), which models preferential-flow discharge as a $\theta_W$-driven threshold process. Another important aspect here is slope flow, that is, the tendency of snow to redistribute meltwater along

layer boundaries for distances of hundreds of meters downhill (Eiriksson et al., 2013; Webb et al., 2018). Both measurements and consequently parametrizations of this process are still very rare, and more work is needed before this process can be included in parsimonious models like S3M.

    Fourth, the conversion from snow to ice in S3M is very simplified, and completely skips the intermediate stage of firn (Cuffey and Paterson, 2010). While we usually turn off the glacier-mass balance when using S3M in flood forecasting, and

while accumulation of multi-year snow as firn will likely characterize only very high elevations in a warming climate, this transition is still an important process to capture from the perspective of physical realism. Considering firn may also extend the applicability of S3M to polar regions, where for example firn-storage capacity is an important factor in determining the long-term fate of the Greenland ice sheet (Forster et al., 2014; Machguth et al., 2016). In this regard, Banfi and De Michele (2021) have recently proposed a local model of snow-firn transition for a binary-mixture snowpack like that considered in S3M.



## 5 Conclusions

We presented S3M v5.1, a spatially explicit hydrology-oriented cryospheric model that successfully reconstructs seasonal snow and glacier evolution through time. The model comprises parametrizations for precipitation-phase partitioning, snow and glacier energy and mass balances, snow rheology and so density evolution, snow aging and albedo, and various provisions for data assimilation. Overall, the model channels elements from the state of the art in cryospheric sciences into a parsimonious and computationally efficient model. Regarding snow, specific elements of relative novelty in this regard are an explicit representation of snow liquid-water content and a hybrid physics-based and temperature-index approach to snowmelt that decouples the radiation- and temperature-driven contributions. The glacier component also includes the well-known $\Delta$h parametrization and the possibility to feed the model with a distributed debris-driven melt factor, both comparatively new approaches in the field. S3M provides an open-source platform to simulate snow and glacier dynamics with the necessary physical realism for hydrologic purposes. Together with the hydrologic model Continuum (Silvestro et al., 2013), S3M fulfills the recurring need for integrated glacio-hydrologic models in both scientific research and operational practice, and can provide the basis for more robust, large scale predictions of the fate of the cryosphere at multiple time scales — from hours to centuries ahead.

*Code availability.* The S3M snow model is available on CIMA Foundation's Hydrology and Hydraulics repository at https://github.com/c-hydro/s3m-dev (GitHub organization), including algorithms to prepare input data and set up computational environments and libraries. S3M is also available on Zenodo at https://doi.org/10.5281/zenodo.4663899. By accessing or using S3M and by extension the Flood-PROOFS modelling system, code, data, and documentation, the user agrees to be bound by the license available at https://github.com/c-hydro/s3m-dev/blob/main/LICENSE.rst. The software is provided "as is", without warranty of any kind, express or implied. In no event shall the authors or copyright holders be liable for any claim, damages, or other liability, whether in an action of contract, tort, or otherwise, arising from, out of, or in connection with the software or the use of other dealings in the software.

*Data availability.* Data used in this paper are available through the Aosta Valley Regional Authority, the Snow and Avalanche Office (Ufficio Neve e Valanghe - RAVDA), and the Aosta Valley Environmental Protection Agency.

*Author contributions.* FA, SG, FD, and FS developed S3M v5.1, with contributions from various colleagues over the course of the last ∼15 years. FA carried out model evaluation in Aosta valley, with inputs from SG, EC, UMdC, SR, and HS. EC and UMdC collected snow-course and glacier data and shared general knowledge about cryospheric processes in the study catchments. SR and HS provided weather and snow data collected by the Aosta Valley Regional Authority, as well as shared general knowledge about hydrologic processes in the study catchments. FA prepared the manuscript, with inputs from all coauthors.

*Competing interests.* Authors declare no competing interest.



*Acknowledgements.* This work was supported by the Aosta Valley Regional Administration, the Italian Civil-Protection Department, and CVA S.p.A. The authors also would like to thank all colleagues who contributed to developing previous versions of S3M over the course of the last ∼15 years. Authors also thanks the Aosta Valley Regional Authority, the Snow and Avalanche Office (Ufficio Neve e Valanghe - RAVDA), and the Aosta Valley Environmental Protection Agency for validation data.

685





**Appendix A: User Manual**

**A1    Run preparation**

S3M v5.1 requires two categories of input, compulsory data: **dynamic**, weather and **static**, topographic inputs. Optionally, the model also ingests **assimilation** data, in the form of either snow-depth and SCA maps or SWE maps (Updating and SWE maps, respectively, see Section 2). The format file required by S3M v5.1 for all inputs is the NetCDF format in the standard GNU zip compression algorithm (gzip, extension .nc.gz).

**A1.1    Static data**

Mandatory static data include:

- a Digital Elevation Model (DEM, in m ASL);

- a raster with metric areas of each computational-grid cell (so-called AreaCell, in $m^2$);

- a glacier mask.

These static rasters must have the same geographic grid and reference system, which will define the computational grid of the model. The glacier mask will indicate which pixels are covered by glaciers, using a unique integer identifier that is passed through the parameter list (so-called Namelist or Infofile, see below for details). S3M v5.1 will use this identifier to select pixels for which glacier melt must be computed *if module G1 is activated* (see Section 2.4). If modules G2 or G3 are selected, S3M v5.1 will use glacier thickness instead (see next paragraph).

Optionally, the user can also provide:

- a glacier-ID raster assigning specific pixels to a glacier according to an inventory;

- a glacier-thickness raster (in m);

- a debris-coefficient raster (see Equation 40);

- a $\Delta h$ pivot table (see Section 2.6).

The glacier-ID raster expects each glacier to be indicated using an integer, and S3M v5.1 will initially compile a list of these unique integers as a glacier inventory for the simulation. If the $\Delta h$ parametrization is chosen, then S3M expects one record of the pivot table for each of these integers. If any of these maps is not supplied, S3M v5.1 will initialize it on the fly using -9999 values, which is S3M's identifier for Not-a-Number values; note that this may lead to inconsistent or erroneous results. S3M v5.1 expects all static input rasters to be included in one single NetCDF file called Terrain_Data.nc.gz (Figure A1). Some basic Python code to generate Terrain_Data.nc.gz is reported at https://github.com/c-hydro/fp-geo-s3m and https://doi.org/10.5281/zenodo.4639614, including a JSON configuration file.





| Terrain_Data.nc.gz | S3M, Static Data | Local File |
|---|---|---|
| AreaCell | AreaCell | Geo2D |
| crs | crs | — |
| GlacierDebris | Glacier debris | Geo2D |
| GlacierID | Glacier ID | Geo2D |
| GlacierMask | GlacierMask | Geo2D |
| Latitude | latitude coordinate | Geo2D |
| Longitude | longitude coordinate | Geo2D |
| PivotTable | PivotTable DeltaH | 2D |
| Terrain | Terrain | Geo2D |
| Thickness | Thickness | Geo2D |

(a) Content of Terrain_Data.nc.gz

| MeteoData_20160901000... | MeteoData VdA | Local File |
|---|---|---|
| AirTemperature | AirTemperature | Geo2D |
| crs | crs | — |
| IncRadiation | IncRadiation | Geo2D |
| Latitude | latitude coordinate | 2D |
| Longitude | longitude coordinate | 2D |
| Rain | Rain | Geo2D |
| RelHumidity | RelHumidity | Geo2D |
| Terrain | geometric height a.s.l. | Geo2D |
| time | time | — |
| Wind | Wind | Geo2D |

(b) Content of MeteoData_yyyymmddHHMM.nc.gz

**Figure A1.** Content of the NetCDF files expected by S3M v5.1 to ingest static and dynamic input data (related to topography and weather, respectively). Note the names of each field, which must be followed in order for S3M v5.1 to load the underlying data. Also note that these NetCDF files include grids with latitude and longitude as well as information regarding time and the reference system. Some Python code that can be adapted to generate these files is reported at https://github.com/c-hydro/fp-hyde. These images were obtained by opening example NetCDF files using Panoply (https://www.giss.nasa.gov/tools/panoply/)

### A1.2 Dynamic data

Mandatory weather input data include:

- air temperature;

- relative humidity;

- incoming shortwave radiation;

- total precipitation.

Weather data must be supplied as distributed raster files according to a common geographic grid and reference system, similar to static data. This geographic grid may in principle be different from the computational grid used by the model, as S3M v5.1 includes a regridding algorithm that automatically checks for concistency and resamples input data using a nearest-neighbor





(a) Content of Updating_yyyymmddHHMM.nc.gz

(b) Content of SWEass_yyyymmddHHMM.nc.gz

**Figure A2.** Content of the NetCDF files expected by S3M v5.1 to ingest assimilation data. Note the names of each field, which must be strictly followed in order for S3M v5.1 to load the underlying data. Also note that these NetCDF files include grids with latitude and longitude as well as information regarding time and the reference system. These images were obtained by opening example NetCDF files using Panoply (https://www.giss.nasa.gov/tools/panoply/)

approach. Note, however, that this nearest-neighbor approach may be unsuitable for specific applications, as for example it
725   does not conserve precipitation mass. The timestep of input data must be the same as the one chosen for model computations.
If any of these weather-input maps is not supplied, S3M v5.1 will initialize it on the fly using -9999 values, which is S3M's
identifier for Not-a-Number values; note that this may lead to inconsistent or erroneous results.

   S3M v5.1 expects weather-input rasters for each time step to be included in one NetCDF file called MeteoData_yyyymmddHHMM.nc.gz,
where yyyymmddHHMM must be replaced with the time-step year (four digits), month, day, hour, and minute (all two digits).
730   Content of one of these files is showed in Figure A1; note that wind fields are not necessary for S3M v5.1. Some code that can
be adapted to prepare input files for S3M v5.1 using Python is available at https://github.com/c-hydro/fp-hyde.

**A1.3   Assimilation data**

Assimilation data are supplied to S3M v5.1 in a similar format as weather data (Figure A2): snow-depth and SCA are bundled
in an Updating_yyyymmddHHMM.nc.gz NetCDF file, while SWE data are supplied in a SWEass_yyyymmddHHMM.nc.gz,
735   where yyyymmddHHMM must be replaced with the time-step year (four digits), month, day, hour, and minute (all two digits).
If assimilation is activated (see next paragraph), S3M v5.1 will look for each of these files every computational timestep; if
any of these files is available for a given timestep, it will be loaded and ingested by the model, otherwise the model will throw





a warning message and simply no assimilation for that time step will be performed. Some initial code that can be adapted to prepare these files for S3M v5.1 using Python is available at https://github.com/c-hydro/fp-hyde.

### A1.4 The Namelist

Besides input and assimilation data, a key step during run preparation is to set up a list of all model options, including paths, modules, and parameter values. In S3M v5.1, this list is referred to as a Namelist or Infofile and is supplied as an ordinary txt file in a pre-defined format. One example of Namelist is available at https://github.com/c-hydro/s3m-dev, while Table A1 details its entries, along with their format, meaning, and options. Further details on parameters and modules can be found in the main text (Section 2). Note that the name of each parameter in the Namelist may be different from notation used in Section 2, mainly because the Namelist reflects definitions used in the source code over the course of ∼15 years of model development. However, comments in the Namelist and details in Table A1 guarantees correspondence with Section 2.

### A2 Run execution

### A2.1 Compiling S3M

S3M v5.1 runs on Linux Debian/Ubuntu 64bit environments, and is expected to run with any other Linux system. On the other hand, no portability to Windows platform is currently possible. The model requires a number of libraries and packages to be pre-installed on the machine, such as the netcdf4 library to handle NetCDF files and a fortran compiler (gnu fortran or Intel fortran) to build the source code. Flood-PROOFS, CIMA Research Foundation's toolkit for hydrologic forecasting, offers a number of shell scripts to set up all required libraries (see https://github.com/c-hydro/fp-envs) and to automatically compile S3M v5.1 (see https://github.com/c-hydro/s3m-dev). The user is strongly recommended to use these pre-existing shell scripts, as they automatically configure all required packages for running the model, perform a number of consistency checks, and allow one to pre-set executable name and properly link the netcdf4 library to model executable. The ReadMe file at https://github.com/c-hydro/s3m-dev explains this set-up phase step by step.

Once all libraries are installed and source code is compiled, S3M v5.1 is launched by storing in a given directory the executable (e.g., S3M_v5_p1.x file) and the Namelist (e.g., S3M_namelist.txt, see Section A1 and Table A1). The user must then point to this directory through the command line and write:

```
$ ./S3M_v5_p1.x S3M_namelist.txt
```

Upon model initialization and throughout model execution, S3M v5.1 will return several pieces of information on the terminal, including as a minimum time-step data and warnings. Note that the user can increase the amount of information reported on the terminal by setting the debugging mode through the Namelist (see Table A1).





## A2.2 The restart file

S3M v5.1 supports reading initial conditions from an external file and use those conditions to restart a simulation. This restart file is in NetCDF format and should include – as a minimum – rasters of SWE, $SWE_D$, $SWE_W$, snow age $A_s$, snow albedo, bulk-dry-snow density $\rho_D$, cumulative daily snowfall and melt, and average air temperature over the previous 1 and 10 days

770 ($\bar{T}_{10d}$). If glacier modules G2 or G3 are activated, S3M will also look for glacier thickness and cumulative annual ice melt (only needed for G3), unless the user has instructed S3M to load glacier thickness from the static-data input file (Table A1).

If any of these rasters is not available, S3M will set them to -9999 (missing values). If the restart option is not activated in the namelist (see Table A1), then S3M will initialize them as appropriate. Because of its nature of forecasting model, the restart file in S3M is simply the relevant output file from a previous simulation; if so, an output file with timestamp 11PM is preferred

775 as it is the most complete output file for that simulation day (see Section A3 for output-file format).

## A3 Run post-processing

Throughout model run, S3M saves NetCDF files with a number of select output variables. The frequency of these output files is chosen by the user through the Namelist (Table A1). Similar to all input files, outputs come in the standard GNU-zip-compression format and are automatically generated with name S3M_yyyymmddHHMM.nc.gz, where yyyymmddHHMM is

780 the time-step year (four digits), month, day, hour, and minute (all two digits).

Figure A3 shows the content of one of these output files, with Table A2 detailing the meaning of each field and how they relate to model variables in Section 2. Field names in the output files are occasionally different from notation used in Section 2, because S3M-output files are used by other models within CIMA's Flood-PROOFS toolkit and so naming strikes a balance across disciplinary jargon, model versions, and legacy with other tools. Note that the list in Figure A3 refers to the extended

785 output mode as defined in the Namelist (Table A1); Table A2 specifies which variables are also saved with a basic output mode.


| S3M_201609012300.nc.gz | S3M_201609012300.nc.gz | Local File |
|---|---|---|
| AgeS | Snow Age | Geo2D |
| AlbedoS | Snow Albedo | Geo2D |
| H_S | Bulk Snow Depth | Geo2D |
| Latitude | Latitude Coordinate | 2D |
| Longitude | Longitude Coordinate | 2D |
| MeltingG | Glacier Melt | Geo2D |
| MeltingS | Snow Melt | Geo2D |
| MeltingSDayCum | Daily Cumulative Snow Melt | Geo2D |
| Outflow | Snowpack Runoff | Geo2D |
| Precip | Total Precipitation Amount | Geo2D |
| RainFall | Rainfall Amount | Geo2D |
| REff | Effective Rainfall | Geo2D |
| RefreezingS | Snow Refreezing | Geo2D |
| Rho_D | Dry Snow Density | Geo2D |
| RhoS | Bulk-Snow Density | Geo2D |
| RhoS0 | Fresh-Snow Density | Geo2D |
| SnowFall | Snowfall Amount | Geo2D |
| SnowfallCum | Daily Cumulative Snowfall | Geo2D |
| SnowMask | Snow Mask | Geo2D |
| SWE | Snow Water Equivalent | Geo2D |
| SWE_D | Dry SWE | Geo2D |
| SWE_W | Wet SWE | Geo2D |
| T_10Days | Average T 10 Days | Geo2D |
| T_1Days | Air Temperature Last 1 Day | Geo2D |
| Theta_W | Bulk Vol. LWC | Geo2D |
| time | time definition of output datasets | — |
| times | times definition of output datasets | — |

**Figure A3.** Content of the S3M NetCDF output file. This image was obtained by opening this NetCDF file using Panoply (https://www.giss.nasa.gov/tools/panoply/)





Table A1: Entries of S3M Namelist, their format, meaning, and options.

| Entry | Format | Meaning | Options |
|---|---|---|---|
| sDomainName | String | Domain name | - |
| iFlagDebugSet | Integer | Flag for debugging | 0 (no) or 1 (yes) |
| iFlagDebugLevel | Integer | Debugging verbosity | 0 to 3 |
| iFlagTypeData_Forcing_Gridded[a] | Integer | MeteoData format | 1 (bin int), 2 (bin dbl), 3 (NetCDF) |
| iFlagTypeData_Updating_Gridded | Integer | Updating format | 1 (bin int), 2 (bin dbl), 3 (NetCDF) |
| iFlagTypeData_Ass_SWE_Gridded | Integer | Ass. SWE format | 1 (bin int), 2 (bin dbl), 3 (NetCDF) |
| iFlagRestart | Integer | Restarting a run[b] | 0 (no) or 1 (yes) |
| iFlagSnowAssim | Integer | Assimilating Updating maps | 0 (no) or 1 (yes) |
| iFlagSnowAssim_SWE | Integer | Assimilating SWE maps | 0 (no) or 1 (yes) |
| iFlagIceMassBalance | Integer | Glacier module | 0 (G1), 1 (G2), 2 (G3) |
| iFlagThickFromTerrData | Integer | Loading glacier thickness from static data[c] | 0 (no) or 1 (yes) |
| iFlagGlacierDebris | Integer | Glacier-debris correction (Eq. 40) | 0 (no) or 1 (yes) |
| iFlagOutputMode | Integer | Output-file format | 0 (basic) or 1 (extended) |
| iFlagAssOnlyPos | Integer | Assimilate only pos. differences | 0 (no) or 1 (yes) |
| a1dGeoForcing | Real | Comma-sep. MeteoData lower-left angle coordinate | - |
| a1dResForcing | Real | Comma-sep. MeteoData cell sizes | - |
| a1iDimsForcing | Integer | Comma-sep. MeteoData dimensions | - |
| iSimLength | Integer | Simulation length in hours | - |
| iDtModel | Integer | Model time-step in seconds | - |
| iDtData_Forcing | Integer | Model time-step in seconds | - |
| iDtData_Output | Integer | Output time-step in seconds | - |
| iDtData_Updating | Integer | Updating time-step in seconds | - |
| iDtData_AssSWE | Integer | Ass. SWE time-step in seconds | - |
| iScaleFactor_Forcing | Integer | MeteoData binary-data scaling factor[a] | - |
| iScaleFactor_Update | Integer | Updating binary-data scaling factor | - |
| iScaleFactor_SWEass | Integer | Ass. SWE binary-data scaling factor | - |
| sTimeStart | String | Timestamp of simulation start | - |
| sTimeRestart | String | Timestamp of restart[b] | - |
| sPathData_Static_Gridded | String | Path to static input data[d] | - |
| sPathData_Forcing_Gridded | String | Path to weather input data[e] | - |
| sPathData_Updating_Gridded | String | Path to Updating data[e] | - |
| sPathData_Output_Gridded | String | Path to output data[e] | - |
| sPathData_Restart_Gridded | String | Path to restart data[b] | - |
| sPathData_SWE_Assimilation_Gridded | String | Path to Ass. SWE data[e] | - |

*Table A1: The table continues in the next page*



*Table A1: Continued from previous page*

| Entry | Format | Meaning | Options |
|---|---|---|---|
| a1dArctUp | Real | Parameter $m_r'$ for four elevation bands[f] | - |
| a1dAltRange | Real | Elevation-band limits for parameter $m_r'$ in m[g] | - |
| iGlacierValue | Integer | Glacier identifier in glacier mask | - |
| dRhoSnowFresh | Real | Maximum fresh-snow density in kg m$^{-3}$ | - |
| dRhoSnowMax | Real | Maximum bulk-snow density in kg m$^{-3}$ | - |
| dRhoSnowMin | Real | Minimum bulk-snow density in kg m$^{-3}$ | - |
| dSnowQualityThr | Real | Snow-quality threshold for assimilation | - |
| dMeltingTRef | Real | Threshold-temperature for melting ($T_\tau$, in °C) | - |
| dIceMeltingCoeff | Real | Ice-melting coefficient [h] | Legacy param., see note and set to 1. |
| iSWEassInfluence | Integer | Number of validity days after SWE-map issue date [i] | |
| dWeightSWEass | Real | Maximum weight $W$ in Equation 34. | |
| dRefreezingSc | Real | Optional multiplicative factor in Equation 19. [j] | See note and set to 1. |
| dModFactorRadS | Real | Parameter $m_{rad}'$ [k] | - |
| sWYstart | String | Water-year starting month (two digits) | - |
| dDebrisThreshold | Real | Threshold-value in $f_{debris}$ to apply Equation 40[l] | - |
| sCommandZipFile | String | Command to zip files | - |
| sCommandUnzipFile | String | Command to unzip files | - |
| sCommandRemoveFile | String | Command to remove files | - |
| dRhoW | Real | Liquid-water density in kg m$^{-3}$ | - |
| sReleaseVersion | String | Model version | - |
| sAuthorNames | String | Authors | - |
| sReleaseDate | String | Release date | - |

[a] S3M v5.1 accepts assimilation and weather input data in binary format (integer or double) in addition to NetCDF. However, this is only allowed for legacy reasons and is discouraged for new applications, so we do not discuss the binary format in this paper.

[b] See Section A2 for details on restarting a run.

[c] If 0 is selected, then S3M will load glacier thickness from a restart file, which is useful when pausing and restarting multi-year simulations.

[d] This must include the full path to the file between single quote marks, without the file name (e.g., '/home/S3M/').

[e] These data can either be stored in one folder, or preferably in a year/month/day directory. In the first case, one should specify here the full path of the folder between single quote marks (e.g., '/home/S3M/data/'). In the second case, one can use automatic directory construction through, e.g., '/home/S3M/data/$yyyy/$mm/$dd/'.

[f] Comma separated.

[g] Three comma-separated values. The four elevation bands are defined as (1) all pixels below first value in a1dAltRange; (2) all pixels between the second and the first values in a1dAltRange; (3) all pixels between the third and second values in a1dAltRange; (4) all pixels above the third value in a1dAltRange.

[h] In previous, unpublished versions of S3M, this parameter was a multiplicative term in Equation 18 to compute a melting parameter for bare ice. This modification was supposed to account for albedo decay on ice, which is now an explicit variable in S3M (see Equation 15). While this parameter is still tunable in the current version of S3M, mainly for legacy reasons, it is recommended to set it to 1 for physical reasons. Accordingly, it was not included in Section 2.

[i] The Kernel function in Equation 34 is set to 0 after this date and the assimilation-SWE map is discarded.

[j] This parameter is an optional, multiplicative term in Equation 19 to reduce $m_r$ in refreezing conditions, similar to Avanzi et al. (2015) or Schaefli et al. (2014). Owing to reasons discussed in Section 2, it is recommended to set this parameter to 1.

[k] Compared to $m_r'$, $m_{rad}'$ cannot be inputed for elevation bands. This will be included in future releases.

[l] Equation 40 is only applied for pixels where $f_{debris}$ is larger than this threshold value.





Table A2: Output-file content (see Section A for details).

| Entry | Model variable | Meaning | Basic mode? | Comments |
|---|---|---|---|---|
| AgeS | $A_s$ | Snow age | Y | - |
| AlbedoS | $\alpha_S$ | Snow albedo | Y | - |
| Cum_WY_MeltingG | $b_a$ | Cum. annual mass balance | Y | Only with modules G2 and G3 |
| H_S | $h_S$ | Bulk-snow depth | N | - |
| Ice_Thickness | $h_G$ | Ice thickness | Y | Only with modules G2 and G3 |
| Ice_Thickness_Change | $\Delta h_G$ | Ice-thickness change | Y | Only with modules G2 and G3 |
| MeltingG | $M_G$ | Glacier melt | N | - |
| MeltingS | $M$ | Snow melt | N | - |
| MeltingSDayCum | - | Cumulative daily snowmelt | Y | Only saved at 11PM |
| Outflow | $O$ | Snowpack runoff | N | - |
| Precip | $P$ | Total precipitation | N | - |
| RainFall | $R_f$ | Rainfall amount | N | - |
| REff | $O + M_G$ | Equivalent precipitation[a] | Y | - |
| RefreezingS | $R$ | Refreezing | N | - |
| Rho_D | $\rho_D$ | Dry bulk-snow density | Y | - |
| RhoS | $\rho_S$ | Bulk-snow density | N | - |
| RhoS0 | $\rho_f$ | Fresh-snow density | N | - |
| SnowFall | $S_f$ | Snowfall amount | N | - |
| SnowfallCum | - | Cumulative daily snowfall | Y | Only saved at 11PM |
| SnowMask | - | Snowmask[b] | Y | - |
| SWE_D | $SWE_D$ | Dry Snow Water Equivalent | Y | - |
| SWE_W | $SWE_W$ | Wet Snow Water Equivalent | Y | - |
| T_10Days | $\bar{T}_{10d}$ | Average 10-day temperature | Y | - |
| T_1Days | - | Average 1-day temperature | Y | Only saved at 11PM |
| Theta_W | $\theta_W$ | Bulk vol. liquid water content | N | - |

[a] Equivalent precipitation is the sum of glacier melt and snowpack runoff (Avanzi et al., 2020a).

[b] Pixels with at least 0.1 mm of SWE (Section 2).



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
