# Peer review of "S3M 5.1: a distributed cryospheric model with dry and wet snow, data assimilation, glacier mass balance, and debris-driven melt"

_Geoscientific Model Development, 2021_

## Referee Comment (RC1)

Review of Avanzi et al.: S3M 5.1: a distributed cryospheric model with dry and wet snow, data assimilation, glacier mass balance, and debris-driven melt

General comments:
The authors present the S3M model, which is a spatial, hydrology oriented cryospheric model using a hybrid physics-based and temperature-index approach. S3M describes seasonal snow and can also account for glacier evolution.

This manuscript presents a detailed model description of the current S3M version (v5.1). All equations, definitions, assumptions, references as well as the required input are thoroughly described. There is a user manual in the Appendix. An evaluation is presented for an inner-Alpine valley (Aosta valley), including calibration and performance analysis.
The manuscript is therefore interesting and valuable for a broad range of scientists and practitioners.
The source code was uploaded on github and is also available on Zenodo (including doi number).

The manuscript is well written and can be easily followed. The authors have done a great job in presenting this comprehensive model suite. I have only a few issues with regards to the model evaluation, which should be addressed before publication. The largest question I have is with regards to the years used to evaluate the model as opposed to the years used to calibrate the model.

Specific comments:

Evaluation years are supposed to be 2004-2009 and 2019 (as indicated in line 496-498). However, in Figure 6 (Section 3.3), evaluation is shown for the calibration years 2013-2019. Figure 9 (line 572-573) also shows a mix of wy used for evaluation consisting of wy used for calibration and those not used for calibration.
Please correct in line 496-498 and so on or/and restrict the evaluation to the non-calibrated years throughout the results sections.

Figure 5d: Was the simulation data for the evaluation with peak-snow-depth courses rounded, or why does this data set has these sharp steps/lines in it? Please explain.

Evaluation at Torgnon study plot (section 3.3): Could you add some performance statistics or mean day differences for reproducing the timing of peak of accumulation and onset/end of ablation?

Figure 7e and Line 552-553: What is the reason that we only see the aspect impact on spatial SWE due to shortwave radiation in one valley in the south of the Rutor glacier? Is it a color bar issue or are there other reasons that this is not visible in the valleys further to the west? If it is a color bar issue, please consider to illustrate the spatial impact differently.

Figure 9d: Should the number of the shown symbols correspond to the number of stake measurements shown in the individual panels? There are only two green dots shown in 9c but 8 blue crosses in 9d (i.e. for the Petit Grapillon)?

Line 595: Maybe presenting the correlation coefficients between $\Delta h_g$ and elevation would be more intuitive?

Why does S3M not scale the diagnostic variables of a grid cell, such as predicted grid cell runoff, with current fractional snow-covered area for that grid cell?

Technical comments:

Eq. (13c): Do you mean $p_s$=1-$p_r$ instead of $p_f$=1-$p_r$ ?

Line 217: Maybe consider changing to: $m_{rad}$ is set to zero if the equation above predicts a negative value.

Line 425: Change "piel" to pixel.

Line 441: Maybe rephrase "..develop for several kilometers.." to "..extend over several kilometers.."?

Figure 7, caption: It might be helpful to indicate for which region the spatial averages are shown. I assume it is the entire region as shown in Fig. 3a.

Line 587ff and Figure 11: Please consider referencing that this glacier is located in Fig. 3c (if it is indeed) or describe its location within the Aosta model region, e.g. in the northern part of Aosta valley.

---

## Author Response (AR1)

Savona (Italy)

January 14, 2022

Dear Dr. Andrew Wickert, Editor,

We would like to submit the manuscript *S3M 5.1: a distributed cryospheric model with dry and wet snow, data assimilation, glacier mass balance, and debris-driven melt* for publication in GMD. The manuscript is a resubmission of manuscript **gmd-2021-92**, which was reviewed by two referees.

We have extensively revised the manuscript based on comments from both referees and would like to thank all of you for finding the time to review our manuscript. We confirm that all requested changes were feasible and we welcomed all of them.

Please find attached our point-by-point replies and the new version of our manuscript for details. We also attached a version of the manuscript with tracked changes.

With our best regards,

*Francesco Avanzi and coauthors*

**Reply to Referee #1**

The authors present the S3M model, which is a spatial, hydrology oriented cryospheric model using a hybrid physics-based and temperature-index approach. S3M describes seasonal snow and can also account for glacier evolution.

This manuscript presents a detailed model description of the current S3M version (v5.1). All equations, definitions, assumptions, references as well as the required input are thoroughly described. There is a user manual in the Appendix. An evaluation is presented for an inner-Alpine valley (Aosta valley), including calibration and performance analysis. The manuscript is therefore interesting and valuable for a broad range of scientists and practitioners. The source code was uploaded on github and is also available on Zenodo (including doi number).

The manuscript is well written and can be easily followed. The authors have done a great job in presenting this comprehensive model suite. I have only a few issues with regards to the model evaluation, which should be addressed before publication. The largest question I have is with regards to the years used to evaluate the model as opposed to the years used to calibrate the model.

> Public response: We thanks Reviewer #1 for their constructive comments. We are happy that the Reviewer appreciated that "all equations, definitions, assumptions, references as well as the required input are thoroughly described", as this was exactly our aim here: writing an extensive and exhaustive description of all aspects of this model, rather than providing only a cursory overview that left many small details hidden in the source code.
>
> All requested revisions are feasible and we will work in this direction as soon as the interactive discussion will be finalized.
>
> Changes to the manuscript: See below.

Evaluation years are supposed to be 2004-2009 and 2019 (as indicated in line 496-498). However, in Figure 6 (Section 3.3), evaluation is shown for the calibration years 2013-2019. Figure 9 (line 572-573) also shows a mix of wy used for evaluation consisting of wy used for calibration and those not used for calibration. Please correct in line 496-498 and so on or/and restrict the evaluation to the non-calibrated years throughout the results sections.

> Public response: This is a very valid point by the Reviewer and we will correct wording regarding Figure 6 as suggested. We will also try to add the rest of the 2020 water year to Figure 6, although data availability for that year is somewhat limited due to the COVID-19 pandemic and the associated restrictions to movement and so field work.
>
> Regarding Figure 9, we will specify that those data were not directly involved in model calibration, although they were taken during water years that were involved in model calibration in terms of snow data.
>
> Changes to the manuscript: The envisaged changes were included in the revised manuscript (see e.g. lines 544f and 630ff).

Figure 5d: Was the simulation data for the evaluation with peak-snow-depth courses rounded, or why does this data set has these sharp steps/lines in it? Please explain.

Public response: Thanks for this question. This is because the spatial resolution of that evaluation dataset is much finer than the spatial resolution of the model (say, ~60 m vs. ~220 m, respectively), as visible in Figure 2c. We performed the evaluation by comparing snow-course data with the concurrently predicted SWE within the same modeling pixel, which means that a number of snow-course data were essentially compared against the same modeling value. This is evident by noting that sharp lines are generally horizontal in Figure 5d, because the same modeled SWE is compared to a variety of observed SWE. We will add these pieces of information to the main text and the caption of Figure 5 for clarity.

Changes to the manuscript: The envisaged changes were included in the revised manuscript (**see the revised caption of Figure 5**).

**Evaluation at Torgnon study plot (section 3.3): Could you add some performance statistics or mean day differences for reproducing the timing of peak of accumulation and onset/end of ablation?**

Public response: Sure, these will be added to the main text as requested.

Changes to the manuscript: The envisaged changes were included in the revised manuscript in the form of Kling Gupta Efficiencies (**see lines 568ff**).

**Figure 7e and Line 552-553: What is the reason that we only see the aspect impact on spatial SWE due to shortwave radiation in one valley in the south of the Rutor glacier? Is it a color bar issue or are there other reasons that this is not visible in the valleys further to the west? If it is a color bar issue, please consider to illustrate the spatial impact differently.**

Public response: Owing to this model including incoming shortwave radiation in addition to air temperature, aspect impact is implicitly included by design, and our development work (in particular that related to the development of the radiation-modulating factor) showed that this impact is correctly implemented. At the same time, a number of factors should be considered when evaluating results in Figure 7, the most important one being that our weather-distribution and snow-assimilation approaches currently augment precipitation and assimilated SWE for two hydropower-relevant catchments in Aosta valley, one of which is that indicated by the Reviewer in their comment. Details on this procedure as well as reasons behind this operational choice are discussed in Avanzi et al. (2021). The result is that spatially distributed SWE for those regions is generally larger than nearby valleys for the same elevation, which may lead to a colorbar issue as the Reviewer noted. While this factor does not affect the overall spirit with which this evaluation is presented (providing "guidelines on how to calibrate S3M in a real-world case study and how to validate and interpret model results for the snow and the glacier component"), we will add this context to the main text and will work on a better figure (and/or rephrase lines 552-553).

Changes to the manuscript: The removed the wording commented by the Reviewer for clarity and brevity (**see lines 604ff**).

**Figure 9d: Should the number of the shown symbols correspond to the number of stake measurements shown in the individual panels? There are only two green dots shown in 9c but 8 blue crosses in 9d (i.e. for the Petit Grapillon)?**

Public response: This is because of repeated measurements taken cross multiple water years as the same location. We will clarify this in the caption of Figure 9.

Changes to the manuscript: The envisaged changes were included in the revised manuscript (**see the revised caption of Figure 9**).

**Line 595: Maybe presenting the correlation coefficients between $\Delta$hg and elevation would be more intuitive?**

Public response: Agreed, this will be included.
Changes to the manuscript: The envisaged changes were included in the revised manuscript (**see line 653**).

**Why does S3M not scale the diagnostic variables of a grid cell, such as predicted grid cell runoff, with current fractional snow-covered area for that grid cell?**

Public response: Thanks for this question. The current assumption in S3M is that simulated SWE and so snowpack runoff are representative of spatially averaged snowpack conditions across the simulated pixel, regardless of *how much* of that pixel is actually covered by snow. This is coherent with, e.g., current practice in many hydrologic models where incoming precipitation is assumed representative of spatially averaged precipitation across the simulated pixel. Having said that, we agree with the Reviewer that adding fractional snow cover and so a full snow-depletion curve (with the associated hysteresis) is an important direction of future development. This will be added to the Discussion section.
Changes to the manuscript: The envisaged changes were included in the revised manuscript (**see lines 175ff**).

**Eq. (13c): Do you mean $p_s = 1 - p_r$ instead of $p_f = 1 - p_r$ ?**

Public response: Correct, this will be fixed.
Changes to the manuscript: The envisaged changes were included in the revised manuscript (**see lines 185ff**).

**Line 217: Maybe consider changing to: m rad is set to zero if the equation above predicts a negative value.**

**Line 425: Change "piel" to pixel.**

**Line 441: Maybe rephrase "..develop for several kilometers.." to "..extend over several kilometers.."?**

Public response: Yes, we will fix these.
Changes to the manuscript: The envisaged changes were included in the revised manuscript (**see e.g. line 250**).

**Figure 7, caption: It might be helpful to indicate for which region the spatial averages are shown. I assume it is the entire region as shown in Fig. 3a.**

Public response: Correct, it is the entire region. We will specify this.
Changes to the manuscript: The envisaged changes were included in the revised manuscript (**see the revised caption of Figure 7**).

**Line 587ff and Figure 11: Please consider referencing that this glacier is located in Fig. 3c**

**(if it is indeed) or describe its location within the Aosta model region, e.g. in the northern part of Aosta valley.**

Public response: Yes, it is part of the considered region (immediately below the North arrow in Figure 3a). We will highlight it in the revised manuscript.

Changes to the manuscript: The envisaged changes were included in the revised manuscript (**see the revised Figure 3**).

**Reply to Reviewer #2**

**In "S3M 5.1: a distributed cryospheric model with dry and wet snow, data assimilation, glacier mass balance, and debris-driven melt" the authors present a distributed cryosphere model to aid in flood forecasting. This is well written and generally easy to follow with a few portions unclear, somewhat due to the manuscript length.**

> Public response: We thank Reviewer #2 for their constructive comments on our draft. All comments are feasible and will be integrated in the revised manuscript, which is now at work. In doing so, we will pay particular attention in better framing model's process representations and novelty in view of the existing literature. We will also revise notation as well as figures as kindly requested. Please find a point by point reply below.

**My main criticism is regarding how the model's process representations and how they are presented. In the abstract the authors note that "Model physics include precipitation-phase partitioning, snow and glacier energy and mass balances, snow rheology and hydraulics, and a data-assimilation protocol." This led me to believe I would be reading a paper about an energy balance model, however this is not at all the case. Indeed the model is a basic temperature-index snow model with a radiation component. There are no internal snowpack energetics, no longwave losses, no turbulent heat fluxes, and no sublimation. Nor is there a vegetation canopy parameterization. Large portions of the study area (c.f. Figure 8) appears to have vegetation cover, so I am at a loss as to how this interaction can be ignored. These are critical components that have been identified by snow hydrologists for many years. [...]**

**I understand that the authors address some of these issues as 'future work', such as the canopy. However, these processes are so critical for basin hydrology that, to present a model to tie into a hydrology model without key cold-region processes, seems unfinished and incomplete. I also don't agree with the framing that t-index parametrizations are a high degree of physical realism, especially when so many other critical processes are omitted. Lastly, I am surprised by the authors stating that coupling radiation to a temperature index model is novel. Hock (2003) details approaches that include this idea and more recent examples exist, e.g., Follum et al (2015).**

> Public response: We thank the reviewer for this insightful comment, which provides us with an opportunity to clarify the contribution of our paper in the context of the existing literature.
>
> First of all, we totally agree with the reviewer that internal snowpack energetics, longwave losses, turbulent heat fluxes, sublimation, and canopy-snow interactions are all important drivers of snowpack evolution. For example, turbulent heat fluxes are among the most significant drivers of snowmelt during rain-on-snow events in alpine regions (Würzer et al., 2016), while previous work like Lundquist et al. (2013) showed that canopy can either accelerate or delay snowmelt based on concurrent climate. Including all these processes in a snowpack model is certainly a key to achieving the most accurate representation of snow temporal dynamics and spatial patterns, especially at fine resolutions. We also agree with the reviewer that solving the full energy balance not only allows one to better track the above processes, but importantly increases model robustness in ungauged conditions and reduces parameter

calibration (although our experience with such models is that some degree of empiricism remains in how to tune or set hard-to-measure model parameters, see Essery et al., 2013).

At the same time, including each of those processes in a model comes with an extra cost in terms of input data requirements, computational resources, and turnaround. While computational resources are quickly increasing, in operational settings there remains a need for critically evaluating what is needed and at what extra cost. In this regard, recent literature has showed that the added value of complex, physics-based snow models over more parsimonious alternatives remains elusive *for variables that are relevant to hydrology.* For example, Magnusson et al. (2015) compared three types of such models and concluded that "errors in the input and validation data, rather than model formulation, seem to be the greatest factor affecting model performance. The three model types provide similar ability to reproduce daily observed snowpack runoff when appropriate model structures are chosen. Model complexity was not a determinant for predicting daily snowpack mass and runoff reliably." Similarly, Avanzi et al. (2016) compared a simple snow model with some parallelisms to S3M with the much more complex Crocus model and found that "the expected loss in performance in the one-layer temperature-index model with respect to the multilayer model is low when considering snow depth, snow water equivalent and bulk snow density." Other two examples include Girons Lopez et al. (2020), who found that "increasing the degree of detail of the temperature-based snow routines in rainfall-runoff models did not necessarily lead to an improved model performance per se. Instead, performing an analysis on which processes are to be included, and to which degree of detail, for a given model and application is a better approach to obtain more reliable and robust results", and Günther et al. (2019), who found that "an improvement in the knowledge (i.e., reduction of uncertainty) of one factor alone" among forcing data errors, model structure, and parameter choices "might not necessarily improve model results". While these findings do not diminish the quest for more complex and holistic models, they do support pragmatic choices when it comes to real-world operations. For example, the Swiss operational snow-hydrological service still maintains a temperature index model in addition to energy-balance models (https://www.slf.ch/en/snow/snow-as-a-water-resource/snow-hydrological-forecasting/oshd-models.htmltabelement1-tab2).

S3M has been developed, validated, and made available with these pragmatic choices in mind. The ultimate goal is to provide a ready-to-use, fully distributed, robust, and parsimonious tool for snow hydrology applications over large areas and when a short turnaround is needed. The model has been under development and use for the last 15 years and is currently being successfully used for snow-hydrology forecasting at various scales, including a national-scale real-time version for civil-protection purposes monitoring snow patterns across Italy at 200 m resolution and with a hourly turnaround.

Compared to standard cryospheric modules in hydrologic models, S3M offers a number of additional features that – in our opinion – motivate publication. These include (1) a separation between the dry and wet phase of snow and thus an explicit representation of snow-wetness patterns across the landscape; (2) the inclusion of glacier dynamics in the form of the state-of-the-art $\Delta$h parametrization and support for debris-modified melt; (3) an approach to snowmelt that fully decouples the temperature- and radiation-driven components. Regarding point (2), we note that there might have been some lack of clarity from our side, particularly in highlighting that S3M is not only "a temperature-index snow model with a radiation component", but also includes glacier processes that are relevant to hydrology but have been only recently made available (Huss et al., 2010; Seibert et al., 2018). Regarding point (3), we agree with the reviewer that including radiation in T-index models is nothing new, as we discussed in the original manuscript (lines 199ff in the original manuscript). What we

found underused in the literature is an explicit separation of the temperature- and radiation-driven components as done by Pellicciotti et al. (2005). In this regard, Hock (1999) embeds potential radiation in the melt factor, with some shortcomings that are discussed in Pellicciotti et al. (2005), while Follum et al. (2015) replaces the temperature term with a proxy from a radiation balance. The latter may be suitable in regions where snowmelt is mostly radiation driven (Bales et al., 2006, e.g., the western US), but would need some form of temperature dependency in temperate regions like the Alps.

Expected changes: We agree with the reviewer that all aspects above should be better and coherently discussed in our manuscript. Some of these arguments are already scattered throughout the draft, but we are planning to consolidate them to better frame our modeling choices – perhaps in the Introduction or at the beginning of the Model description section. In so doing, we will give proper credits to all processes discussed by the reviewer.

Changes to the manuscript: The envisaged changes regarding model novelty and philosophy were included in the revised manuscript (**see lines 84ff**). In doing so, we also revised the Abstract to clarify our contribution (**see lines 7ff**). We also added more context regarding novelty behind the explicit separation of the temperature- and radiation-driven component in the melt computation of S3M (**see lines 232ff**).

**Certainly this criticism could be waved away as a differing in model philosophy, however the results have issues that suggest there is something quite wrong with these simplifications as applied. In Figure 6 there are multiple years (2013, 2014, 2015, 2016) and sometimes multiple occurrences per year, where 1m to 2m snowpacks are almost instantly ablated. I don't understand the physical process by which this could occur. Rain on snow run a-muck? A warm, sunny day "melting" a deep snowpack that has no cold content tracked? I understood these results to be with the data assimilation turned on (this was a bit unclear to me). Assuming this is true, then without the data assimilation system these results would have been even more wrong. The DA is then massively compensating for broken parameterizations. If this was without DA, then how is it ablating a 1m snow pack, then immediately reestablishing a 1m snowpack? A similar situation occurs on the glacier, where 2x melt is predicted or the case of Petit Graphillon where multiple meters of ablation are observed with no reaction of the model. Considering this model is described as a flood forecasting model, I am concerned with the non-physical behaviour that is being exhibited here.**

Public response: We apologize for our lack of clarity here. Those abrupt oscillations in simulated snow depth and SWE are due to the assimilated dataset, which is the operational result of a multiregression model fitted across observed snow depth at ultrasonic-sensor stations and a number of physiographic features (Avanzi et al., 2021). These regressions often maintain – or even propagate – measurement noise, a frequent issue of ultrasonic snow-depth sensors (Ryan et al., 2008). This is demonstrated by the revised Torgnon figure (see Figure 1 below), which also reports open loop simulations for snow depth and SWE (red, dashed). These open loop simulations do not display the same abrupt oscillations as seen in assimilated simulations. Thus, these oscillations are not due to potential broken parametrizations. In this regard, note that S3M v5.1 has been subject to intense debugging prior to release, including various mass and consistency checks.

Regarding the Petit Grapillon: we only have one year of data for that glacier (summer 2012), where glacier surveys were then discontinued because of extensive and deep crevasses that make point measurements non-representative of the actual melt pattern (see Figure 2

below). Besides, melt patterns for this glacier are accelerated by frequent avalanches due to high surface slopes, which are currently not included in S3M. Both these factors represent limitations that should be taken into consideration when interpreting results for the Petit Grapillon.

Expected changes: We will add all the above to the manuscript, as well as replace Figure 6 with Figure 1 below.

Changes to the manuscript: The envisaged changes were included in the revised manuscript (**see lines 574ff and the revised Figure 6, as well as lines 635ff**).

[Figure]

**Figure 1**: *Comparison between measurements and modeling outputs at the Torgnon study plot. First, second, third, and fourth columns are snow depth (HS), SWE, bulk-snow density, and albedo, respectively. The third and fourth columns also report dry-snow density and bulk volumetric liquid-water content, respectively. Each row is one water year (2013 to 2020).*

[Figure]

***Figure 2***: *The Petit Grapillon glacier, showing the pronounced crevasses challenging ablation measurements on this glacier.*

**Page 1, L4 a high degree of physical realism: Do you mean they are mostly empirical? Certainly many models have good physical representations. Indeed I'd expect most models that simulate physical processes to have physical realism!**

> Expected changes: Agreed, we will rephrase.
> Changes to the manuscript: Done (**see line 4**).

**L7 reconstructs: Simulates?**

> Expected changes: Agreed, we will rephrase.
> Changes to the manuscript: Done (**see line 6**).

**L16 the paper comprises an user manual: I would like to see an actual science question. In my opinion, even in GMD, there should be some hypothesis testing and scientific questions answered that support model development.**

> Public response: We agree with the reviewer that science questions are an essential component of research papers. However, we followed guidelines of GMD when drafting our

manuscript as a Model description paper, that is a "comprehensive descriptions of numerical models which fall within the scope of GMD." (see `https://www.geoscientific-model-development.net/about/manuscript_types.html#item1`). According to these guidelines, "the main paper should describe both the underlying scientific basis and purpose of the model and overview the numerical solutions employed. The scientific goal is reproducibility: ideally, the description should be sufficiently detailed to in principle allow for the re-implementation of the model by others, so all technical details which could substantially affect the numerical output should be described." This is why we did not formulate specific research questions for this manuscript.

Expected changes: We discussed this comment with the Topical Editor. While adding specific hypothesis testing would result in both significant restructuring and likely an increase in lenght of the paper, we will be more explicit on identifying important scientific questions that could be solved using S3M.

Changes to the manuscript: We discussed this comment among coauthors and found it difficult to locate specific passages that could succinctly be revised in this sense. On the other hand, some passages in the Discussion section already provide questions that could be answered using S3M (**see lines 660ff**). For this reason, and in agreement with the spirit of a Model description paper, we are proposing not to include additional text on this matter for the sake of brevity (this is already a quite long manuscript).

**Page 2, L22 during the warm, summer season: And spring**

Expected changes: Agreed, we will rephrase.
Changes to the manuscript: Done (**see line 21**).

**L22 when demand:By whom?**

Expected changes: We meant water demand by human societies and ecosystems. This will be clarified.
Changes to the manuscript: Done (**see line 21**).

**L25 while 1.4+ billion people in Asia rely on discharge from high-mountain: Citation?**

Public response: Citation is Immerzeel et al. (2010), as reported at the end of the line. No change.

**L33 large portfolio: W/c**

Expected changes: This will be rephrased.
Changes to the manuscript: Done (**see line 32**).

**L36 avalanche forecasting: perhaps "Avalanche hazard forecast"**

Changes to the manuscript: Done – here and elsewhere (**see line 35**).
Expected changes: This will be rephrased.

**L36 so weather: ?**

Expected changes: This will be rephrased.
Changes to the manuscript: Done (**see line 35**).

**L37 aridity: AR6 suggests location dependent, consider citing the newest IPCC report**

> Expected changes: This will be rephrased.
> Changes to the manuscript: Done (**see line 37**).

**L42 Regarding seasonal snow: Suggest adding FSM (Essery, 2015) to this list.**

> Expected changes: Essery (2015) will be added.
> Changes to the manuscript: Done (**see line 45**).

**Page 3, L60 The evidence that simplified and complex models often yield comparable predictive: Certainly many models exist that show better SWE and sd when including full physics e.g Lafaysse (2017) and Vionnet (2021). Even when considering just SWE and SD, the inclusion of multi-layer snowpack models is important for deep mountain snow covers, or rain on snow events.**

**L 63 low complexity when it comes to internal layering and micro-scale properties: Ok but what about the other processes that impact?**

**L65 real-world, L67 these four factors trace back: As written this suggests obs are related to empiricism, and I don't understand how Obs location biases are due to empiricism.**

**L71 all the four factors: Seems like factors is used as "requirement" which is not true.**

**L72 parsimonious as for: Suggest this isn't a requirement**

> Public response: Please see our reply to the first main comment above on these important matters, which are all somewhat related to parsimony in model representation vs. physical realism of model predictions. We iterate that we totally agree with the reviewer that internal snowpack energetics, longwave losses, turbulent heat fluxes, sublimation, and canopy-snow interactions are all important drivers of snowpack evolution. Including all these processes in a snowpack model is certainly a key to achieving the most accurate representation of snow temporal dynamics and spatial patterns, especially at fine resolutions. At the same time, including each of those processes in a model comes with an extra cost in terms of input data requirements, computational resources, and turnaround. While computational resources are quickly increasing, in operational settings there remains a need for critically evaluating what is needed and at what extra cost. In this regard, recent literature has showed that the added value of complex, physics-based snow models over more parsimonious alternatives *for variables that are relevant to hydrology* remains elusive (Rutter et al., 2009; Magnusson et al., 2015; Zaramella et al., 2019; Girons Lopez et al., 2020; Günther et al., 2019). While these findings do not diminish the quest for more complex and holistic models, they do support pragmatic choices when it comes to real-world operations. S3M has been developed, validated, and made available with these pragmatic choices in mind. The ultimate goal is to provide a ready-to-use, fully distributed, robust, and parsimonious tool as required by snow hydrology applications over large areas and when a short turnaround is needed (this was the intended meaning of "requirements" in this context).
> Expected changes: We will reformulate this passage to clarify both applicability and modeling philosophy, as well as the focus on operational applications.
> Changes to the manuscript: The envisaged changes regarding model novelty and philosophy were included in the revised manuscript (**see lines 84ff**). In doing so, we also revised the

Abstract to clarify our contribution (**see lines 7ff**). We also added more context regarding novelty behind the explicit separation of the temperature- and radiation-driven component in the melt computation of S3M (**see lines 232ff**).

**L75 and so: ?**

Expected changes: we will rephrase in "a spatially explicit prediction of both dry and wet-snow spatial patterns, AS WELL AS bulk snowpack liquid water content.
Changes to the manuscript: Done (**see line 72**).

**L75 avalanche forecasting: Without the microstructure is this really true?**

Public response: We agree with the reviewer that avalanche forecasting necessarily requires more complex, multi-layer models. This is why we framed S3M as a hydrology-oriented cryospheric model. At the same time, Mitterer et al. (2013) suggested an index defined as the average liquid water content of the entire snowpack as a potential predictor of wet snow avalanches, while we have direct experience of avalanche forecasters taking into account regional-scale predictions by S3M in their assessment of avalanche risk.
Expected changes: we will use the above wording to clarify this passage.
Changes to the manuscript: We added Mitterer et al. (2013) as an additional reference for the sake of brevity (**see line 73**).

**L81 Section 3 presents an example of results for an inner alpine valley: Ok great! I would like to see a science question. Even GMD should have-science questions and hypothesis testing**

Public response: We agree with the reviewer that science questions are an essential component of research papers. However, we followed guidelines of GMD when drafting our manuscript as a Model description paper, that is a "comprehensive descriptions of numerical models which fall within the scope of GMD." (see `https://www.geoscientific-model-development.net/about/manuscript_types.html#item1`). According to these guidelines, "the main paper should describe both the underlying scientific basis and purpose of the model and overview the numerical solutions employed. The scientific goal is reproducibility: ideally, the description should be sufficiently detailed to in principle allow for the re-implementation of the model by others, so all technical details which could substantially affect the numerical output should be described." This is why we did not formulate specific research questions for this manuscript.
Expected changes: We discussed this comment with the Topical Editor. While adding specific hypothesis testing would result in both significant restructuring and likely an increase in lenght of the paper, we will be more explicit on identifying important scientific questions that could be solved using S3M.
Changes to the manuscript: We discussed this comment among coauthors and found it difficult to locate specific passages that could succinctly be revised in this sense. On the other hand, some passages in the Discussion section already provide questions that could be answered using S3M (**see lines 660ff**). For this reason, and in agreement with the spirit of a Model description paper, we are proposing not to include additional text on this matter for the sake of brevity (this is already a quite long manuscript).

**Page 4: L86 being**

Public response: This passage seems grammatically correct to the authors.
Expected changes: No change.

**L86 no spatial interdependency: I read this to mean no lateral mass or energy transfer? Perhaps be explicit**

Public response: Correct.
Expected changes: This will be clarified.
Changes to the manuscript: Done (**see line 105**).

**L90 using a forward-Euler method: This is a basic solver, why this method versus the RK, BE, etc methods? What tolerances were used? I assume a constant step size?**

Public response: All equations are ordinary differential equations with no need for iterative computations. Thus we used a simple forward-Euler method as it provides comparatively high numerical stability and minimizes computational time. The time step of the model is flexible, although our experience is mainly with hourly modeling setups, in agreement with the usual frequency of ground network observations. Variables are defined as either integer or double precision 8-byte real numbers.
Expected changes: This will be clarified.
Changes to the manuscript: Done (**see lines 108ff**).

**L106 The density of glacier ice is assumed equal to i: So no further compression?**

Public response: Correct.
Expected changes: This will be clarified.
Changes to the manuscript: Done (**see lines 128ff**).

**Page 5, L123 state variable: Some of these look like fluxes e.g MG. The SWE etc looks to be a diagnostic variable and the ice lattice is the actual state variable. Later in the manuscript SWE is noted as a diagnostic variable. I would suggest tidying this up and making the diagram clear.**

Public response: We agree.
Expected changes: Figure 1 will be revised as recommended.
Changes to the manuscript: Done.

**L123 inputs: Unclear what is an input: is an albedo input?**

Public response: Inputs are precipitation, air temperature, relative humidity and incoming shortwave radiation (see the user manual).
Expected changes: Figure 1 will be revised as recommended.
Changes to the manuscript: Done.

**L128 where $S\hat{}$: What do the hats denote? I don't think it is noted explicitly in the text**

Public response: $S\hat{}$ is a mass flux in kg/$\Delta$t, while e.g. S is in mm/$\Delta$t.
Expected changes: This will be clarified.
Changes to the manuscript: Done (**see lines 152ff**).

**L129 and Oˆ is the outflow mass flux: Above you call this a state in figure 1**

Public response: Oˆ is a mass flux in kg/$\Delta$t.
Expected changes: Figure 1 will be revised as recommended.
Changes to the manuscript: Done.

**Page 6, Figure 1: Main definitions: Add units, see above note on fluxes/state**

Public response: Agreed.
Expected changes: Figure 1 will be revised as recommended.
Changes to the manuscript: Done.

**Page 7, L145 S W ED and S W EW: Are these not diagnostic?**

Public response: No, $SWE_D$ and $SWE_W$ are prognostic variables, wheres e.g. $SWE$ is diagnostic: $SWE = SWE_D + SWE_W$.
Expected changes: This will be clarified.
Changes to the manuscript: Done (**see lines 167ff**).

**Page 8, L169 is standard in degree-day model: From the intro I was not expecting this to be yet another temperature index model**

Expected changes: We will revise the introduction and the abstract to clarify this.
Changes to the manuscript: Done (**e.g., see lines 8ff**).

**L176 seems yielding satisfactory results: Grammar**

Expected changes: This will be revised.
Changes to the manuscript: Done (**see line 205**).

**L176 especially in suppressing mid-winter melt episodes that do not appear in validation data: Mid winter melt will increase with climate warming though. Further, your results show non-physical mid winter ablation events. Lastly this appears to be an ad hoc calibration, is that true?**

Public response:
We agree that mid winter melt will increase with climate warming. Here, we are referring to erroneous mid-winter melt episodes that appear in simulations but DO NOT appear in validation data.
We also agree that our approach to cold content estimation is empirical and may be subject to calibration.
Please refer to our answer at page 3 above regarding the non-physical mid winter ablation events.
Expected changes: All information above will be reported in the manuscript.
Changes to the manuscript: Done (**see line 204**).

**L177 decoupling radiative forcing: I mean, this is the point of a full energy balance model**

Public response:

Please refer to our answer at page 2-3 above regarding this point.

**L181 where Sr is incoming shortwave: This section needs units. I see the below section has it, please put these into the above text**

Expected changes: We agree. We will move units from the below paragraph to this paragraph.

Changes to the manuscript: Done (**see lines 210ff**).

**L184 sets Sr to 0 between 7PM and 7AM according to forcing timestamps: This seems arbitrary. Why not just compute load sun rise and set?**

Public response: We agree that this is a pragmatic decision, with the only aim of removing spurious noise in the radiation data.

Expected changes: We will mention the above and suggest that computing load sun rise and set would provide more realistic results.

Changes to the manuscript: Done (**see lines 218ff**).

**Page 9, L195 otherwise: Move to start of statement**

Expected changes: Agreed.

Changes to the manuscript: Done.

**L197 (timestamp time): What does this mean?**

Public response: This is a typo, apologies.

Expected changes: This will be removed.

Changes to the manuscript: Done.

**L200 sensitivity of S3M to both is rather low: Based on?**

Public response: This statement is based on results of our application in Aosta valley as well as general experience with this model.

Expected changes: We will clarify this passage based on the above wording.

Changes to the manuscript: Done (**see lines 229ff**).

**L201 closer to physics: What does this mean? Do you mean closer to a fully first-principals energy balance model?**

Public response: Correct.

Expected changes: This will be rephrased as suggested.

Changes to the manuscript: Done (**see lines 230ff**).

**L202 this hybrid approach: There are tons of temp+rad formulations. Either this isn't new or the contribution is not clear me to me**

Public response:
See our reply at page 2-3 on this important matter.

Expected changes: We will rephrase as noted at page 2-3.

Changes to the manuscript: Done (**see lines 232**).

**L 206 Which is an isothermal, very efficient condition for shortwave radiation to convert into actual melt: Awkward, suggest clarify**

Public response: Agreed.

Expected changes: We will rephrase as "This isothermal condition is very efficient for shortwave radiation to convert into actual melt"

Changes to the manuscript: Done (**see line 239**).

**L208 regardless of the actual cold content: Is the case the authors making that their contribution is a cold content temp index? Cold content is /required/ to correctly track energetics in deep mountain snowcovers**

Public response: Our empirical approach to melt suppression based on a temperature-modulated efficiency parameter is a modification to the original approach by Pellicciotti et al. (2005). As such, it is a novel aspect in the realm of enhanced temperature index methods.

Expected changes: We will clarify this point in the manuscript.

Changes to the manuscript: Done (**see lines 244ff**).

**L 210 To mimic this transition: This is what the cold content tracking should do, sn't it? It is not clear to me how this approach works with deep mountain snowcovers.**

**Page 10, L229 Refreezing is computed: Does this refreezing latent heat flux decrease the cold content ie warm snowpack?**

Public response: Our temperature-modulated efficiency parameter is a proxy of cold content, but it does not imply an *explicit* computation of cold content. It is a first attempt to take into account thermal inertia in subfreezing conditions and how it is related to external climate, similarly to Schaefli and Huss (2011). At the present stage, no relation with snow depth or internal snow temperature is included, but we agree that this could be an helpful addition.

Expected changes: The above will be added to the manuscript.

Changes to the manuscript: Done (**see lines 251ff**).

**L230 eq 19, R: Remind the reader what "R" is please**

Expected changes: This is refreezing (line 229). We will specify this.

Changes to the manuscript: Done (**see line 265**).

**Page 12, L269 Report instabilities of Equation 20: Could this be due to using FE? And to be clear, this is the use of 20 that is a problem and not the solution to 20?**

**L 269 high saturation value: So numerics are likely a problem?**

**L269 very shallow snowpack: This is classically a tough problem and requires a good eb model + good numerical scheme**

Public response: There was lack of clarity from our side here, and particularly the choice of wording like "instabilities" was misleading. What we meant here is simply that high saturation

values in a shallow snowpack may lead to large outflow rates and non-physical negative values of $SWE_W$. So outflow is limited to concurrent $SWE_W$ in such situations.

We also agree with the reviewer that shallow and/or highly saturated snowpacks are challenging settings, although their relevance for hydrologic predictions in alpine contexts is limited.

Expected changes: We will clarify as outlined above.

Changes to the manuscript: Done (**see lines 304ff**).

**L 276 a representative element at 66% depth: Where does 66% come from? This seems arbitrary.**

Public response: Assuming a linear profile for stress with depth (which is realistic in static conditions), an element at 66% depth experiences an average stress. This is why it is called a representative element here.

Expected changes: We will clarify as outlined above.

Changes to the manuscript: Done (**see lines 313ff**).

**Page 13, L290 While it is set to 0°C otherwise: So a 2m mountain snow cover is set to instantly isothermal? Perhaps I am misreading this, but it seems to me that the authors are suggesting that as soon as Tair ¿0, they set the ground temperature to be = Tair? If that's what is happening, then that is completel wrong, especially for deep mountain snow covers. If that isn't what is happening, then please clarify this, as despite reading it a few times I am still uncertain on what, exactly, is being done.**

Public response: We agree with the reviewer that this passage needs clarifications. Snow mean temperature as parametrized in S3M for settling predictions has no relation with snow melt, and so with the thermal inertia of subfreezing snowpacks. The parametrization of snow mean temperature is needed only because the snow settling equation used by S3M includes this variable as a predictor, following early pieces of evidence that snow settling rate depends on temperature (Mellor, 1975). The fact that this variable has no implication for snow melt, compounded by the sensitivity of the settling equation to snow temperature being rather small (not reported for brevity) led us to introducing this simple parametrization here. In this context, previous work has showed that (1) soil temperature in alpine conditions remains at ∼0°C throughout winter (Ohara and Kavvas, 2006; Filippa et al., 2014), and thus (2) a linear profile for temperature with depth is plausible, at least for a simple, first-order approximation like the one we are using here (De Michele et al., 2013). We agree that this is a simplification that should be better explained and spelled out.

Expected changes: We will clarify as outlined above.

Changes to the manuscript: Done (**see lines 326ff**).

**L293 thus implying that refreezing has no impact on snow structure: So, what is the point then? Latent heat and Cold Content tracking? I am picking on this specifically due to the noting of avalanche hazard forecasting in the introduction**

Public response: The point is to correctly simulate the dynamics of interstitial liquid water, particularly as it regards the simulation of bulk liquid water content and snowpack runoff. Previous work (De Michele et al., 2013; Avanzi et al., 2015) has showed that including refreezing is a prerequisite for capturing liquid water content during melt freeze cycles. Meanwhile,

the effects of refreezing on snow structure (e.g., on specific surface area) are unclear and were not parametrized.

Expected changes: We will clarify as outlined above.

Changes to the manuscript: This sentence was removed for clarity.

**Section 3.5 Data assimilation: This is direct insertion, correct? I understand the authors not wanting to cut and paste verbatim from existing papers, and I appreciate them keeping the length of this manuscript down. However I did struggle through this section to know, exactly, how this was done. Specifically that Swe and Sd are diagnostic variables, but seem to be assimilated as a state variable**

Public response: Newtonian Relaxation (also known as nudging) is different from direct insertion. In direct insertion, model estimates are replaced by observations; in a nudging scheme, the correction factor is proportional to the difference between observations and model outputs via a Kernel weight (Boni et al., 2010; Mazzoleni et al., 2018). Equation 36 is the essence of this nudging scheme, while lines 340ff details how prognostic variables are updated.

Expected changes: We will clarify as outlined above.

Changes to the manuscript: Done (**see lines 377ff**).

**Page 14, L 307 of Updating: Fix cap**

Expected changes: Agreed.

Changes to the manuscript: The cap was maintained as Updating is a proper noun in this context.

**Page 15, L347 total SWE in S3M v5.1 is only a diagnostic variable: As noted above this needs to be fixed in figure 1 and the text throughout for consistency**

Expected changes: Agreed.

Changes to the manuscript: Done (**see lines 168ff**).

**L347 S3M also supports assimilating only positive differences in Equation 36, that is, only correcting modeled SWE if observations are larger than simulations: I am a bit surprised by this tactic. Certainly it helps recover from the otherwise catastrophic mid winter melts, but doesn't help with over estimates in SWE/SD. I'd like to see a bit more elaboration as to why it is done this way**

Public response: Assimilation of positive differences is a pragmatic choice of S3M that is common to other operational models, such as the Swiss SNOWPACK (Bartelt and Lehning, 2002; Lehning et al., 2002b,a). The basic idea is to assume snow depth changes as a proxy of snowfall events: assimilating positive differences will allow the model to correct well known issues of precipitation gauges such as precipitation undercatch (Ryan et al., 2008) while preventing assimilation data from perturbing the reconstruction of the snowmelt period. This is an optional setting though, and one can run S3M by assimilating both positive and negative differences. We iterate that the so-called "catastrophic mid winter melts" were not due to model parametrizations, but instead noise in the assimilation dataset.

Expected changes: We will clarify as outlined above.

Changes to the manuscript: Done (**see lines 389ff**).

**Page 16, L376 equivalent to G1: Maybe section 2.4.1 would benefit from text that notes what g1 is as I got here and was confused. G1 is only used in the heading. Would be maybe nice to remind the reader what this is.**

Expected changes: Agreed.
Changes to the manuscript: Done (**see line 406**).

**Page 17, L 391 any residual SWE at the end of each water year is added to hG: Ok so this ignores firn then. In multi-year firn processes I'm skeptical this can be ignored, but I understand that in some Alps glaciers this approximation can be valid. I would like to see supporting literature for this assumption.**

Public response: We agree that the inclusion of firn is an important future step, as we mention in the Discussion section. In this regard, previous work by Schaefli et al. (2005) in Switzerland showed that "for the analysed hydro-climatic area, the use of a separate degree day factor for firn does not improve neither the discharge nor the mass balance simulation". While this result may be specific to high-elevation Alpine catchments, and/or related to model overparametrization (Schaefli et al., 2005), it does raise the issue of whether including firn as an additional component has a significant impact on model results compared to just considering snow and ice – especially in a warming climate. In general, this intermediate component between snow and ice has been rarely included in intermediate-complexity cryospheric and/or hydrologic models.
Expected changes: We will clarify as outlined above.
Changes to the manuscript: Done (**see lines 438ff**).

**L 398 parametrizing: Spelling**

Expected changes: Agreed.
Changes to the manuscript: Done (**see line 444**).

**Page 18, L 446 spatialize and downscale weather-input data: Aren't these data already spatial datasets? So is this just a downscale to the numerical model grid? Please note the methods used to do so (eg what spatial interpolant/regridder is being used) [ ah, I see there is a note in the appendix on this, perhaps either reduce the text and move it into the main body or explicitly note the appendix section].**

Expected changes: We will refer to the Appendix section, as S3M does not include spatialization or downscaling techniques and this goes beyond the scopes of this paper.
Changes to the manuscript: Done (**see lines 493ff**). Note that the Appendix was moved to the Supporting Information.

**L 448 using the Continuum model: What is this?**

Public response: This is the hydrologic model used in Aosta valley in combination with S3M.
Expected changes: We will clarify this.
Changes to the manuscript: We removed these unnecessary details.

**Page 20, L470 the notions that: This is unclear to me what you mean by notions here. Do**

**you mean 'notion'? 100% seems very high**

> Public response: We meant "the fact that".
> Expected changes: We will rephrase this.
> Changes to the manuscript: Done (**see lines 516ff**).

**L470 of transmitted shortwave radiation: Transmitted through what? No reflectance?**

> Public response: We meant trasmitted through snow, after computing the reflected component.
> Expected changes: We will clarify this.
> Changes to the manuscript: Done (**see line 517**).

**L476 mainly for computational-resource constraints: Less brute force methods can be quite efficient. E.g., Saman Razavi, Razi Sheikholeslami, Hoshin V. Gupta, Amin Haghnegahdar, VARS-TOOL: A toolbox for comprehensive, efficient, and robust sensitivity and uncertainty analysis, Environmental Modelling Software, Volume 112, 2019**

> Public response: We agree, although the aim of this section was not to discuss calibration protocols but rather to show an example of model use, in line with GMD guidelines.
> Expected changes: We will add this.
> Changes to the manuscript: Done (**see lines 521ff**).

**L486 , in line with expectations, L489 that the sensitivity of S3M to mrad and mr is surprisingly low: Doesn't this go against what was previously stated? If this is the case though, then what is causing the huge mid winter ablation events?**

> Public response: We see no contradiction here: optimal values in model parameters are quite robust throughout years, which we interpret as due to the decoupling between the temperature and the radiation components bringing S3M closer to a first-principals energy balance model. As already noted above, mid winter abrupt changes in snow depth were not due to model parametrizations, but instead to noise in the assimilation dataset.
> Expected changes: We will add the above to the manuscript.
> Changes to the manuscript: Done (**see lines 538ff**).

**Page 21, 3.2 Evaluation: point snow depth: Should note this has assimilation in it is my thinking was that this was sans assimilation like the previous section.**

> Public response: This was already noted at line 498. No change.

**L 498 Sections**

> Expected changes: We will change with "sections".
> Changes to the manuscript: Done.

**Page 23, 3.3 Evaluation: the Torgnon study plot: This is with assimilation right? I'm not totally sure why but I'm struggling to remember which runs have assimilation and which do not.**

Public response: Yes, it includes assimilation (see line 498).

Expected changes: We will iterate this concept, as well as add open-loop simulations as per our reply to the comment above at page 3.

Changes to the manuscript: Done (**see legend in Figure 6**).

**L519 (Figure 6): What is going on with the mid winter ablation events? It seems assimilation is heavily compensating for these impacts?**

Public response: See our reply to the comment above at page 3.

**L536 not reported for brevity**

Public response: This comment remained unclear to us, so no change is envisaged.

**L536 qualitatively: Wouldn't this be quantitatively?**

Public response: We did not compute formal biases or other performance metrics, so this remains a qualitative statement. No change.

**Page 27 L562 provides renovated: Not sure what this means. W/c**

Public response: We meant "new".

Expected changes: We will rephrase.

Changes to the manuscript: Done (**see line 616**).

**L566 S3M is among the first parsimonious snow models to provide such information: I don't accept that temp index models can work when applied to climate change forecasts. The loss of stationarity eg Milly(2008) leads to huge challenges in applying hindcast calibrated models to future conditions. For example, the increase in mid winter ablation makes calibrated models that assume a spring melt (and implicitly calibrate for the deep snowpacks) will likely fail when applied to very different types of winters.**

Public response: We agree that applying calibration-based models to changing conditions is challenging, although literature evidence on this matter is still sparse. In this passage, we did not imply robustness in this sense. Rather, we are simply making the point that S3M provides spatially explicit predictions of snow liquid water content and these patterns are increasingly important in a warming climate.

Expected changes: We will rephrase based on the above.

Changes to the manuscript: Done (**see lines 620ff**).

**L 576 solid results: Word choice**

Expected changes: We will replace with " Simulations using the G3 glacier module returned a correlation ..."

Changes to the manuscript: Done (**see line 632**).

**L580 and the Petit Grapillon: These results seem really poor if observations show 2m of ablation but 0m is predicted!**

Public response: See our reply at page 3 on this.

**Page 29, L600 at a comparatively high standard in physical realism: A temperature index snowmelt model with no: canopy interactions, sublimation, blowing snow, energy balance, etc. I am not convinced this is a high amount of physical realism.**

> Public response: We agree that this passage needs rephrasing. One should always keep in mind that our benchmark here are cryospheric modules of operational hydrologic models, which rarely include all the above processes and very rarely include a state of the art representation of glacier dynamics.
>
> Expected changes: We will rephrase as above.
>
> Changes to the manuscript: Done (**see lines 657ff**).

**Page 30 L610 to produce future scenarios of: See my previous comment on applying calibrated tindex models to future climates.**

> Public response: See our reply above.

**L613 for an ordinary laptop: What does this mean? Please give a sense of CPU arch, speeds, etc. As a model design philiosophy, I am not so sure that constraining the physics and conceptualization so-as to run on a laptop is optimal.**

> Expected changes: We will rephrase as indicated. Note that minimizing model computational requirements while allowing landscape scale predictions does represent an asset for regions of the world or operational services that cannot afford a multicore server.
>
> Changes to the manuscript: Done (**see line 671**).

**L616 the scarcity of open- source suites reconnecting them with the hydrologic cycle: I don't I understand this comment**

> Expected changes: We agree. This passage will be deleted for clarity.
>
> Changes to the manuscript: Done.

**Page 32 L658 energy and mass balances: Temperature index models are not an energy balance.**

> Expected changes: We will rephrase as recommended.
>
> Changes to the manuscript: Done.

**L679 I've never seen a license agreement in a code availability section. Not saying it is wrong, but I personally think it should be removed, indeed the code itself should have the license agreement in it.**

> Expected changes: The GitHub repository of S3M includes the same license agreement (`https://github.com/c-hydro/s3m-dev/`), so this will be moved to the Appendix.
>
> Changes to the manuscript: Done (**see lines 736ff**).

**L715I see that this is where the input data requirements are. I think this should be referred to explicitly from the main text**

Expected changes: Agreed, we will do so.

Changes to the manuscript: Done (**see lines XXf**).

**L751 I was not able to get this to compile on Macos due to a unlimit being specified in the .sh**

Public response: S3M supports Linux Debian/Ubuntu 64bit environments (lines 750ff), and we have no experience with Macos.

Expected changes: We will specify this. We will also test our setup script on a Macos machine and try make it compatible with such systems.

Changes to the manuscript: This is now specified in the Supporting Information.

**References**

Avanzi, F., De Michele, C., Morin, S., Carmagnola, C.M., Ghezzi, A., Lejeune, Y., 2016. Model complexity and data requirements in snow hydrology: seeking a balance in practical applications. Hydrological Processes .

Avanzi, F., Ercolani, G., Gabellani, S., Cremonese, E., Pogliotti, P., Filippa, G., Morra di Cella, U., Ratto, S., Stevenin, H., Cauduro, M., Juglair, S., 2021. Learning about precipitation lapse rates from snow course data improves water balance modeling. Hydrology and Earth System Sciences 25, 2109–2131. URL: https://hess.copernicus.org/articles/25/2109/2021/, doi:10.5194/hess-25-2109-2021.

Avanzi, F., Yamaguchi, S., Hirashima, H., De Michele, C., 2015. Bulk volumetric liquid water content in a seasonal snowpack: modeling its dynamics in different climatic conditions. Advances in Water Resources 86, 1 – 13. doi:10.1016/j.advwatres.2015.09.021.

Bales, R., Molotch, N.P., Painter, T.H., Dettinger, M.D., Rice, R., Dozier, J., 2006. Mountain hydrology of the western United States. Water Resources Research 42, W08432. doi:10.1029/2005WR004387.

Bartelt, P., Lehning, M., 2002. A physical SNOWPACK model for the Swiss avalanche warning Part I: numerical model. Cold Regions Science and Technology 35, 123–145. doi:10.1016/S0165-232X(02)00074-5.

Boni, G., Castelli, F., Gabellani, S., Machiavello, G., Rudari, R., 2010. Assimilation of modis snow cover and real time snow depth point data in a snow dynamic model, in: 2010 IEEE International Geoscience and Remote Sensing Symposium, pp. 1788–1791.

De Michele, C., Avanzi, F., Ghezzi, A., Jommi, C., 2013. Investigating the dynamics of bulk snow density in dry and wet conditions using a one-dimensional model. The Cryosphere 7, 433–444. doi:10.5194/tc-7-433-2013.

Essery, R., 2015. A factorial snowpack model (fsm 1.0). Geoscientific Model Development 8, 3867–3876. URL: https://gmd.copernicus.org/articles/8/3867/2015/, doi:10.5194/gmd-8-3867-2015.

Essery, R., Morin, S., Lejeune, Y., Ménard, C.B., 2013. A comparison of 1701 snow models using observations from an alpine site. Advances in Water Resources 55, 131–148. doi:10.1016/j.advwatres.2012.07.013.

Filippa, G., Maggioni, M., Zanini, E., Freppaz, M., 2014. Analysis of continuous snow temperature profiles from automatic weather stations in Aosta Valley (NW Italy): Uncertainties and applications. Cold Regions Science and Technology 104 - 105, 54 – 62.

Follum, M.L., Downer, C.W., Niemann, J.D., Roylance, S.M., Vuyovich, C.M., 2015. A radiation-derived

temperature-index snow routine for the gssha hydrologic model. Journal of Hydrology 529, 723–736. URL: `https://www.sciencedirect.com/science/article/pii/S0022169415006599`, doi:`https://doi.org/10.1016/j.jhydrol.2015.08.044`.

Girons Lopez, M., Vis, M.J.P., Jenicek, M., Griessinger, N., Seibert, J., 2020. Assessing the degree of detail of temperature-based snow routines for runoff modelling in mountainous areas in central europe. Hydrology and Earth System Sciences 24, 4441–4461. URL: `https://hess.copernicus.org/articles/24/4441/2020/`, doi:10.5194/hess-24-4441-2020.

Günther, D., Marke, T., Essery, R., Strasser, U., 2019. Uncertainties in snowpack simulations—assessing the impact of model structure, parameter choice, and forcing data error on point-scale energy balance snow model performance. Water Resources Research 55, 2779–2800. URL: `https://agupubs.onlinelibrary.wiley.com/doi/abs/10.1029/2018WR023403`, doi:`https://doi.org/10.1029/2018WR023403`, arXiv:`https://agupubs.onlinelibrary.wiley.com/doi/pdf/10.1029/2018WR023403`.

Hock, R., 1999. A distributed temperature-index ice- and snowmelt model including potential direct solar radiation. Journal of Glaciology 45, 101–111.

Huss, M., Jouvet, G., Farinotti, D., Bauder, A., 2010. Future high-mountain hydrology: a new parameterization of glacier retreat. Hydrology and Earth System Sciences 14, 815–829. URL: `https://www.hydrol-earth-syst-sci.net/14/815/2010/`, doi:10.5194/hess-14-815-2010.

Immerzeel, W.W., van Beek, L.P.H., Bierkens, M.F.P., 2010. Climate Change Will Affect the Asian Water Towers. Science 328, 1382–1385.

Lehning, M., Bartelt, P., Brown, B., Fierz, C., 2002a. A physical SNOWPACK model for the Swiss avalanche warning Part III: meteorological forcing, thin layer formation and evaluation. Cold Regions Science and Technology 35, 169–184.

Lehning, M., Bartelt, P., Brown, B., Fierz, C., Satyawali, P., 2002b. A physical SNOWPACK model for the Swiss avalanche warning Part II. Snow microstructure. Cold Regions Science and Technology 35, 147–167.

Lundquist, J.D., Dickerson-Lange, S.E., Lutz, J.A., Cristea, N.C., 2013. Lower forest density enhances snow retention in regions with warmer winters: A global framework developed from plot-scale observations and modeling. Water Resources Research 49, 6356–6370. doi:10.1002/`wrcr`.20504.

Magnusson, J., Wever, N., Essery, R., an A. Winstral, N.H., Jonas, T., 2015. Evaluating snow models with varying process representations for hydrological applications. Water Resources Research 51, 2707 – 2723.

Mazzoleni, M., Noh, S.J., Lee, H., Liu, Y., Seo, D.J., Amaranto, A., Alfonso, L., Solomatine, D.P., 2018. Real-time assimilation of streamflow observations into a hydrological routing model: effects of model structures and updating methods. Hydrological Sciences Journal 63, 386–407. URL: `https://doi.org/10.1080/02626667.2018.1430898`, doi:10.1080/02626667.2018.1430898, arXiv:`https://doi.org/10.1080/02626667.2018.1430898`.

Mellor, M., 1975. A review of basic snow mechanics, in: Snow Mechanics Symposium (Proceedings of the Grindelwald Symposium) April 1974, International Association of Hydrological Sciences Publication, vol. 114, 251? 291.

Mitterer, C., Techel, F., Fierz, C., Schweizer, J., 2013. An operational supporting tool for assessing wet-snow avalanche danger, in: International Snow Science Workshop Grenoble - Chamonix Mont-Blanc - 2013.

Ohara, N., Kavvas, M.L., 2006. Field observations and numerical model experiments for the snowmelt process at a field site. Advances in Water Resources 29, 194–211. doi:`10.1016/j.advwatres.2005.03.016`.

Pellicciotti, F., Brock, B., Strasser, U., Burlando, P., Funk, M., Corripio, J., 2005. An enhanced temperature-index glacier melt model including the shortwave radiation balance: development and testing for haut glacier d'arolla, switzerland. Journal of Glaciology 51, 573–587. doi:`10.3189/172756505781829124`.

Rutter, N., Essery, R., Pomeroy, J., Altimir, N., Andreadis, K., Baker, I., Barr, A., Bartlett, P., Boone, A., Deng, H., Douville, H., Dutra, E., Elder, K., Ellis, C., Feng, X., Gelfan, A., Goodbody, A., Gusev, Y., Gustafsson, D., Hellström, R., Hirabayashi, Y., Hirota, T., Jonas, T., Koren, V., Kuragina, A., Lettenmaier, D., Li, W.P., Luce, C., Martin, E., Nasonova, O., Pumpanen, J., Pyles, R.D., Samuelsson, P., Sandells, M., Schädler, G., Shmakin, A., Smirnova, T.G., Stähli, M., Stöckli, R., Strasser, U., Su, H., Suzuki, K., Takata, K., Tanaka, K., Thompson, E., Vesala, T., Viterbo, P., Wiltshire, A., Xia, K., Xue, Y., Yamazaki, T., 2009. Evaluation of forest snow processes models (snowmip2). Journal of Geophysical Research: Atmospheres 114. doi:`https://doi.org/10.1029/2008JD011063`.

Ryan, W.A., Doesken, N.J., Fassnacht, S.R., 2008. Evaluation of Ultrasonic Snow Depth Sensors for U.S. Snow Measurements. Journal of Atmospheric and Oceanic Technology 25, 667–684. doi:`10.1175/2007JTECHA947.1`.

Schaefli, B., Hingray, B., Niggli, M., Musy, A., 2005. A conceptual glacio-hydrological model for high mountainous catchments. Hydrology and Earth System Sciences 9, 95–109. URL: `http://www.hydrol-earth-syst-sci.net/9/95/2005/`, doi:`10.5194/hess-9-95-2005`.

Schaefli, B., Huss, M., 2011. Integrating point glacier mass balance observations into hydrologic model identification. Hydrology and Earth System Sciences 15, 1227–1241. URL: `https://www.hydrol-earth-syst-sci.net/15/1227/2011/`, doi:`10.5194/hess-15-1227-2011`.

Seibert, J., Vis, M.J.P., Kohn, I., Weiler, M., Stahl, K., 2018. Technical note: Representing glacier geometry changes in a semi-distributed hydrological model. Hydrology and Earth System Sciences 22, 2211–2224. URL: `https://www.hydrol-earth-syst-sci.net/22/2211/2018/`, doi:`10.5194/hess-22-2211-2018`.

Würzer, S., Jonas, T., Wever, N., Lehning, M., 2016. Influence of initial snowpack properties on runoff formation during rain-on-snow events. Journal of Hydrometeorology doi:`http://dx.doi.org/10.1175/JHM-D-15-0181.1`.

Zaramella, M., Borga, M., Zoccatelli, D., Carturan, L., 2019. Topmelt 1.0: a topography-based distribution function approach to snowmelt simulation for hydrological modelling at basin scale. Geoscientific Model Development 12, 5251–5265. URL: `https://gmd.copernicus.org/articles/12/5251/2019/`, doi:`10.5194/gmd-12-5251-2019`.

---

## Referee Report (RR1)

15.05.2022

Dear Francesco Avanzi et al.,

Dear Andrew Wickert,

in the revised manuscript version Avanzi et al. have addressed the review comments that Anonymous Reviewer #1 and Anonymous Reviewer #2 have raised in the first assessment of the manuscript. After my assessment in this second round of reviews, I think the main points of criticism have already been discussed and addressed adequately by the authors.

Having said this, I have just some minor comments I recommend to account for before the manuscript is published. Most of them are of a technical nature, and due to the fact that some statements seem vague or contain "flavored language" (=judgmental/subjective terms). However, overall there are no major statements left that I would particularly disagree with. I am fine with not seeing the manuscript again before publication.

All the best wishes!

In the following, my line numbers refer to the revised submission of the manuscript:

**Structural/content comments**

**l. 218 f.**, and other places: there are a lot of announcements what could be improved in the model and what is planned future work. This is interesting, but it is not typical "paper style" to mention it distributed in the entire manuscript. Comments like these are content I would expect mostly in the outlook section.

**Section 2.5**: Maybe I missed it, but where does the initial glacier thickness/outlines come from? I think this should be cited as well.

**l. 494**: "240m": I think this is the first time where the resolution is mentioned in the text. Shouldn't it occur in a place somewhat more prominent?

**l. 643**: I am confused here about the reason you give for the positive elevation changes at the head of the glacier: in the definition of $h_G$ in l.419 you say that $h_G$ is glacier thickness. The glacier thickness defined in this way and its change shown on Fig. 10 bottom right thus includes surface height (change) including ice, firn and snow changes. $\Delta h_G$ might be positive due to the snow distribution pattern on the glacier, or differing elevation data acquisition dates within the hydrological year. However, according to me the positive change is not due to the snow-to-ice conversion (It would be negative, since snow is compacted when turning into ice!).

**Conclusions**: What I am missing here is a summary of the validation with some numbers that prove how well the model did. You are talking about the fact that this model "channels elements from the state of the art in cryospheric sciences into a parsimonious and computationally efficient model", so, since the manuscript spends quite some paragraphs on validation, a summary of these quality measures should

be found here again (e.g. the efficiencies you have added as response to Reviewer#1's comment on "Evaluation at Torgnon study plot (section 3.3))").

**Technical/grammar/spelling/language comments**

l.**15**: "a user manual"
l.**30**: "about snow-glacier amount": What is meant by this? The amount of snow on glaciers?
l. **34** "applicationS"
l. **47**: please add citation from complex flow approaches, e.g. Jouvet & Huss (2019): Future retreat of Great Aletsch Glacier, Journal of Glaciology.
l. **80**: "Alpine"
l. **204** ff.: "Our approach is the result of intensive trial and error and yields satisfactory results" -> IMHO it is too early to state this here.
l. **213**, **215**: THE model time step
l. **501/505**: "in principle, a fairly large amount" together with "fine-tuned based on expert knowledge" and "our experience suggests": these are four vague expressions concatenated, which leaves me with a feeling of insecurity. My advice: name the exact (amount of) parameters that can be calibrated using standard (e.g. least squares) methods, and those that have to be guessed. If your experience suggests two main calibration parameters, this statement should be supported by how you come to this conclusion (e.g. by a sensitivity analysis), especially in view of the fact that you deny their strong influence afterwards (see also comment on l200/486/489 of Reviewer#2).
l. **516**: "based on preliminary tuning": what exactly is meant by this?
l **548/549**: "note that snow-depth assimilation maps are only generated between August and April in an effort to unaffect the simulation of the depletion phase of the seasonal snowpack." What does this mean?
**Fg. 5**: the position of the panel labels is such that they can really be overlooked easily. I would put them outside of the individual panels in the top left corner.
l. **561**:"complicate"
l. **566**: "satisfactorily": leave out (judgmental)
**Fig.6**: I would add "OL" spelled out, at least in the caption. It should be self-evident what abbreviations in the figures mean without having to dig into the text.
l. **596**: delete "well"
l.**600**: unclear to me what you mean by "warm" and "cold" storm
**Fig.8**: the LWC levels on the colorbar are somewhat arbitrary: why don't you go for a continuous colormap?
l. **627**: This sentence is not understandable to me.
l.**629**: "2012 data" – source?
l.**632**: "satisfactory": here again: I would state only the correlation result and let the reader judge
l.**634**: What does "visually higher" mean?
l. **634 ff**.: no "the" before the proper nouns
l. **636**: first-> former
**Fig. 10/11**: it's basically impossible to compare the individual figure panels, since the colorbar changes its limits constantly. Please ensure they are the same for all panels.

l. **663**: "When is a…"

l. **670 f.**: "a few hours" -> since you give so precise details about the technical equipment, I would have expected a number here, accordingly. (as a consequence of reply to comment on l.613 by Reviewer #2)

l. **668**: "attractive" -> I would use "particular", or similar

l. **672**: "is a factor here": a question arises for me: a factor with which influence? Is netCDF fast or slow compared to what is possible? How much time does it consume with respect to the overall runtime?

l. **676**: delete "such"

l. **680**: "urgent": if you say it like this, it gives the impression you should have done it before publishing this article. Do you mean "help to resolve rain on snow events in the model"?

---

## Author Response (AR2)

Savona (Italy)

June 7, 2022

Dear Dr. Andrew Wickert, Editor,

We would like to submit the manuscript *S3M 5.1: a distributed cryospheric model with dry and wet snow, data assimilation, glacier mass balance, and debris-driven melt* for publication in GMD. The manuscript is a re-resubmission of manuscript **gmd-2021-92**, which was reviewed by a third referee.

We have extensively revised the manuscript based on comments from the referee and would like to thank them for finding the time to review our manuscript.

Please find attached our point-by-point replies and the new version of our manuscript for details. We also attached a version of the manuscript with tracked changes.

With our best regards,

*Francesco Avanzi and coauthors*

**Reply to Referee #3**

**In the revised manuscript version Avanzi et al. have addressed the review comments that Anonymous Reviewer #1 and Anonymous Reviewer #2 have raised in the first assessment of the manuscript. After my assessment in this second round of reviews, I think the main points of criticism have already been discussed and addressed adequately by the authors. Having said this, I have just some minor comments I recommend to account for before the manuscript is published. Most of them are of a technical nature, and due to the fact that some statements seem vague or contain "flavored language" (=judgmental/subjective terms). However, overall there are no major statements left that I would particularly disagree with. I am fine with not seeing the manuscript again before publication. All the best wishes!**

> We thank Reviewer #3 for their positive assessment of our manuscript.

**l. 218 f., and other places: there are a lot of announcements what could be improved in the model and what is planned future work. This is interesting, but it is not typical "paper style" to mention it distributed in the entire manuscript. Comments like these are content I would expect mostly in the outlook section.**

> We thank the referee for this comment, and have revised the manuscript to avoid such statement whenever possible (e.g., see lines **438ff**). Note that we did keep some of these statements, especially when such clarifications were asked by previous referees (e.g., see lines **438ff or 95ff**), or when they represented only brief and local comments that would be out of context with the content of the Outlook section (e.g., see lines **219-220**).

**Section 2.5: Maybe I missed it, but where does the initial glacier thickness/outlines come from? I think this should be cited as well.**

> We added examples of such datasets in the literature (see lines **424ff**). In this regard, note that Section 2 is a general description of S3M, with no specific reference to the Aosta valley case study.

**l. 494: "240m": I think this is the first time where the resolution is mentioned in the text. Shouldn't it occur in a place somewhat more prominent?**

> We added information in Section 2 regarding the spatial resolution being at user's discretion, while 240 m is just the resolution of our example of use in Aosta valley (see lines **108ff**). In this regard, note that Section 2 is a general description of S3M, with no specific reference to the Aosta valley case study.

**l. 643: I am confused here about the reason you give for the positive elevation changes at the head of the glacier: in the definition of hG in l.419 you say that hG is glacier thickness. The glacier thickness defined in this way and its change shown on Fig. 10 bottom right thus includes surface height (change) including ice, firn and snow changes. $\Delta$ hG might be positive due to the snow distribution pattern on the glacier, or differing elevation data acquisition dates within the hydrological year. However, according to me**

the positive change is not due to the snow-to-ice conversion (It would be negative, since snow is compacted when turning into ice!).

> We apologize for the confusion here. Positive changes in glacier thickness are possible due to the instantaneous conversion of seasonal snow into ice at the end of each water year. This conversion does include compaction, but may still represent a positive term due to residual snow at the end of the water year in areas of minimal melt. We added this clarification to the manuscript (see lines **642ff**).

**Conclusions: What I am missing here is a summary of the validation with some numbers that prove how well the model did. You are talking about the fact that this model "channels elements from the state of the art in cryospheric sciences into a parsimonious and computationally efficient model", so, since the manuscript spends quite some paragraphs on validation, a summary of these quality measures should be found here again (e.g. the efficiencies you have added as response to Reviewer1's comment on "Evaluation at Torgnon study plot (section 3.3))").**

> We agree and added the requested information to the Conclusions (see lines **725ff**).

**l.15: "a user manual"**

**l.30: "about snow-glacier amount": What is meant by this? The amount of snow on glaciers?**

**l. 34 "applicationS"**

**l. 47: please add citation from complex flow approaches, e.g. Jouvet Huss (2019): Future retreat of Great Aletsch Glacier, Journal of Glaciology.**

**l. 80: "Alpine"**

**l. 204 ff.: "Our approach is the result of intensive trial and error and yields satisfactory results" -¿ IMHO it is too early to state this here.**

**l. 213, 215: THE model time step**

> All these technical/grammar/spelling/language comments were included in the revised manuscript.

**l. 501/505: "in principle, a fairly large amount" together with "fine-tuned based on expert knowledge" and "our experience suggests": these are four vague expressions concatenated, which leaves me with a feeling of insecurity. My advice: name the exact (amount of) parameters that can be calibrated using standard (e.g. least squares) methods, and those that have to be guessed. If your experience suggests two main calibration parameters, this statement should be supported by how you come to this conclusion (e.g. by a sensitivity analysis), especially in view of the fact that you deny their strong influence afterwards (see also comment on l200/486/489 of Reviewer2).**

> We agree and rephrased ad recommended (see lines **501ff**). Our focus on the two melt parameters was the result of both manual tuning of parameter values during model development (not a formal sensitivity analysis) and a review of existing literature on this matter

(Hock, 2003; Pellicciotti et al., 2005).

**l. 516: "based on preliminary tuning": what exactly is meant by this?**

We removed this unclear statement and clarified the passage (see lines **515ff**).

**l 548/549: "note that snow-depth assimilation maps are only generated between August and April in an effort to unaffect the simulation of the depletion phase of the seasonal snowpack." What does this mean?**

This means that assimilation was NOT performed between May and July, in order not to perturb the model simulation of snowmelt (see lines **548ff**).

**Fg. 5: the position of the panel labels is such that they can really be overlooked easily. I would put them outside of the individual panels in the top left corner.**

**l. 561:"complicate"**

**l. 566: "satisfactorily": leave out (judgmental)**

**Fig.6: I would add "OL" spelled out, at least in the caption. It should be self-evident what abbreviations in the figures mean without having to dig into the text.**

**l. 596: delete "well"**

All these technical/grammar/spelling/language comments were included in the revised manuscript.

**l.600: unclear to me what you mean by "warm" and "cold" storm**

We were referring to the fact that the first storm corresponded to a rise in air temperature and so was predominantly rainfall-driven, while the second was mostly snowfall-driven. This is a minor detail that was not relevant to the main story, so we removed it for clarity (see lines **595ff**).

**Fig.8: the LWC levels on the colorbar are somewhat arbitrary: why don't you go for a continuous colormap?**

Done.

**l. 627: This sentence is not understandable to me.**

Sentence removed (see lines **622ff**).

**l.629: "2012 data" – source?**

Source added (see lines **628ff**).

**l.632: "satisfactory": here again: I would state only the correlation result and let the reader judge**

l.634: What does "visually higher" mean?

l. 634 ff.: no "the" before the proper nouns

l. 636: first-¿ former

> All these technical/grammar/spelling/language comments were included in the revised manuscript.

**Fig. 10/11: it's basically impossible to compare the individual figure panels, since the colorbar changes its limits constantly. Please ensure they are the same for all panels.**

> We understand referee's point here, but think that the main point of these figures is to compare spatial patterns within each panel, and how they match the $\Delta$h parametrization, rather than spatial patterns across panels (see lines **641ff**). Thus no change was made.

l. 663: "When is a..."

l. 670 f.: "a few hours" -¿ since you give so precise details about the technical equipment, I would have expected a number here, accordingly. (as a consequence of reply to comment on l.613 by Reviewer 2)

l. 668: "attractive" -¿ I would use "particular", or similar

l. 672: "is a factor here": a question arises for me: a factor with which influence? Is netCDF fast or slow compared to what is possible? How much time does it consume with respect to the overall runtime?

l. 676: delete "such"

l. 680: "urgent": if you say it like this, it gives the impression you should have done it before publishing this article. Do you mean "help to resolve rain on snow events in the model"?

> All these technical/grammar/spelling/language comments were included in the revised manuscript. Regarding l672, we meant that reading and saving files is the most significant contributor to computational time. We have not run specific tests on this, so it remains a qualitative information that we think will be important for readers.

**References**

[revised manuscript text omitted]